# Plasmid-mediated metronidazole resistance in *Clostridioides difficile*

Ilse M. Boekhoud [1,2,3], Bastian V.H. Hornung[1,4], Eloisa Sevilla [5], Céline Harmanus [1], Ingrid M.J.G. Bos-Sanders[1], Elisabeth M. Terveer [1], Rosa Bolea [5], Jeroen Corver [1], Ed J. Kuijper[1,3,4,6] & Wiep Klaas Smits [1,2,3]*

Metronidazole was until recently used as a first-line treatment for potentially life-threatening *Clostridioides difficile* (CD) infection. Although cases of metronidazole resistance have been documented, no clear mechanism for metronidazole resistance or a role for plasmids in antimicrobial resistance has been described for CD. Here, we report genome sequences of seven susceptible and sixteen resistant CD isolates from human and animal sources, including isolates from a patient with recurrent CD infection by a PCR ribotype (RT) 020 strain, which developed resistance to metronidazole over the course of treatment (minimal inhibitory concentration [MIC] = 8 mg L$^{-1}$). Metronidazole resistance correlates with the presence of a 7-kb plasmid, pCD-METRO. pCD-METRO is present in toxigenic and non-toxigenic resistant ($n = 23$), but not susceptible ($n = 563$), isolates from multiple countries. Introduction of a pCD-METRO-derived vector into a susceptible strain increases the MIC 25-fold. Our finding of plasmid-mediated resistance can impact diagnostics and treatment of CD infections.

[1] Department of Medical Microbiology, Leiden University Medical Center, Albinusdreef 2, PO Box 9600, 2300 RC Leiden, The Netherlands. [2] Centre for Microbial Cell Biology, Leiden, The Netherlands. [3] Netherlands Centre for One Health, Leiden, The Netherlands. [4] Center for Microbiome Analyses and Therapeutics, Leiden University Medical Center, Leiden, The Netherlands. [5] Departamento de Patología Animal, Facultad de Veterinaria, Universidad de Zaragoza, Miguel Servet 177, 50013 Zaragoza, Spain. [6] National Institute for Public Health and the Environment, Bilthoven, The Netherlands. *email: W.K.Smits@lumc.nl

Clostridioides difficile (Clostridium difficile) is a Gram-positive obligate anaerobe capable of causing Clostridioides difficile Infection (CDI) upon disruption of the normal intestinal microbiota[1]. Although it is one of the major causes of nosocomial infectious diarrhea, community-acquired CDI is becoming more frequent[2,3]. CDI infection poses a significant economic burden with an estimated cost at €3 billion per year in the European Union and impairs the quality of life in infected individuals[4,5]. The incidence of CDI has increased over the last two decades with outbreaks caused by epidemic types such as PCR ribotype (RT) 027 (NAP1/BI)[6]. CDI is not restricted to this type, however, as infections caused by RT001, RT002, RT014/020, and RT078 are frequently reported in both Europe and the United States[7,8]. Metronidazole is used for the treatment of mild-to-moderate infections and vancomycin for severe infections, though vancomycin is increasingly indicated as a general first-line treatment[9–16]. Fidaxomicin has recently also been approved for CDI treatment, but its use is limited by high costs[12]. Fecal microbiota transplantation (FMT) is effective at treating recurrent CDI (rCDI) that is refractory to antimicrobial therapy[17]. Reduced susceptibility and resistance to clinically used antimicrobials, including metronidazole, has been reported and this, combined with the intrinsic multiple drug-resistant nature of C. difficile, stresses the importance for the development of better diagnostics and new effective treatment modalities[8].

Routine antimicrobial susceptibility testing is generally not performed for C. difficile and consequently, reports of resistance to metronidazole are rare[18–20]. Longitudinal surveillance in Europe found that 0.2% of clinical isolates investigated were resistant to metronidazole[19], but reported rates from other studies vary from 0 to 18.3%[21–24]. These differences may reflect geographic distributions in resistant strains, or differences in testing methodology and breakpoints used[25,26]. Moreover, metronidazole resistance can be unstable, inducible and heterogeneous[27]. Finally, metronidazole resistance appears to be more frequent in non-toxigenic strains such as those belonging to RT010, which have a 7–9-fold increase in Minimal Inhibitory Concentration (MIC) values compared to RT001, RT027 and RT078[21,26].

Metronidazole is a 5-nitroimidazole prodrug that upon intracellular reductive activation induces cellular damage through nitro-radicals[27]. It is not only used in the treatment of CDI, but also an important drug for treating parasitic infections and as prophylactic antimicrobial in for instance abdominal surgery[27,28]. Mechanisms associated with metronidazole resistance described in other organisms include the presence of 5-nitroimidazole reductases (nim genes), altered pyruvate-ferredoxin oxidoreductase (PFOR) activity and adaptations to (oxidative) stress[27]. The knowledge on resistance mechanisms in C. difficile is very limited, but may involve modulation of core metabolic and stress pathways as well[29,30]. Of note, levels of metronidazole achieved in the colon are generally low and this could be relevant for the selection of resistant strains[31].

Here, we present a case of a patient with rCDI due to an initially metronidazole susceptible (MTZ$^S$) RT020 strain, which developed resistance to metronidazole over time. We analyze the genome sequences of these toxigenic MTZ$^S$ and metronidazole-resistant (MTZ$^R$) strains, together with 5 MTZ$^S$ and 11 MTZ$^R$ non-toxigenic RT010 strains. We identify pCD-METRO, a 7-kb plasmid conferring metronidazole resistance. This plasmid is internationally disseminated and also occurs in epidemic types. We thus report a clinically relevant phenotype associated with plasmid carriage in C. difficile.

## Results

**In-patient development of a metronidazole-resistant strain.** A 54-year-old kidney–pancreas transplant patient with a medical history of Type I diabetes mellitus, vascular disease and a double lower-leg amputation was on hemodialysis when developing diarrhea. The patient was subsequently diagnosed with CDI and a toxigenic metronidazole sensitive (MIC = 0.25 mg L$^{-1}$) RT020 strain was isolated from the fecal material of the patient. Treatment with metronidazole was started, leading to initial resolution of the symptoms (Fig. 1). Two more episodes of CDI occurred during which the patient was treated primarily with vancomycin (but also metronidazole) prior to an FMT provided by the Netherlands Donor Feces Bank. At the start of the second episode a MTZ$^S$ RT020 strain was once more isolated.

Three months after the first FMT, the patient once again developed bloody diarrhea and two more episodes of rCDI were diagnosed, which were treated with a vancomycin and a

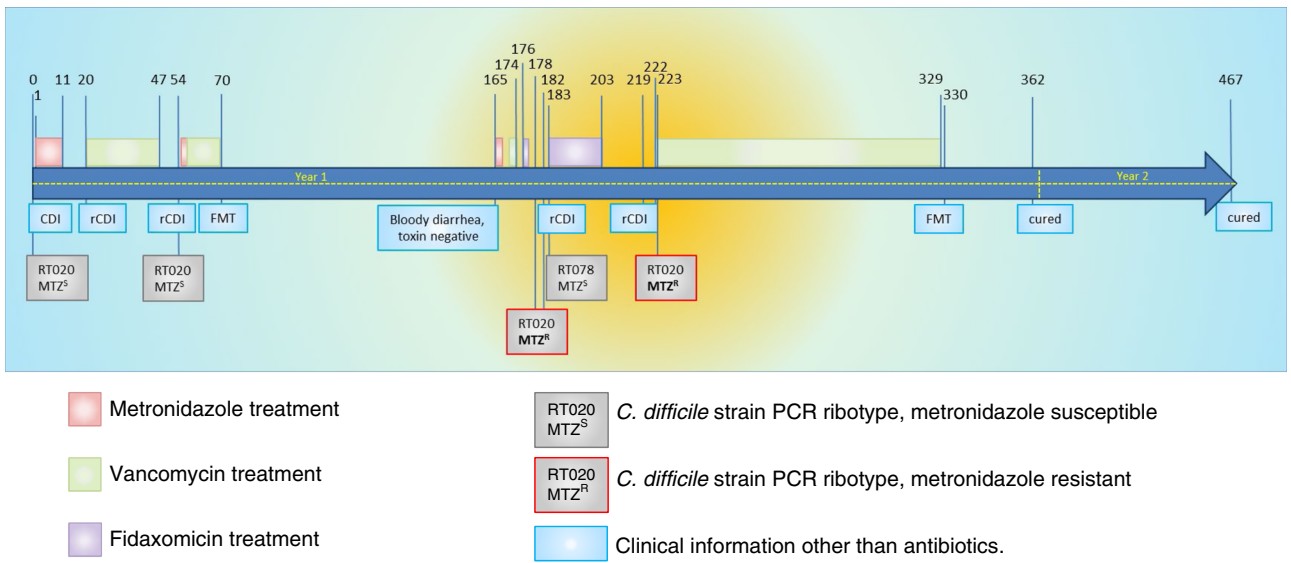

**Fig. 1 Timeline of the course of the antibiotic treatment and rCDI in the patient.** Dates and timepoints on which treatment was initiated or stopped and C. difficile isolates were recovered are indicated above the timeline. (r)CDI was diagnosed when both a toxin enzyme-immune assay and nucleic acid amplification test were positive for C. difficile in combination with a physician's assessment of symptoms consistent with CDI. Yellow highlighting indicates the time where pCD-METRO-positive MTZ$^R$ C. difficile was isolated. Source data are provided as a Source Data file.

fidaxomicin regime. At two instances, RT020 strains were again isolated from the fecal material of the patient. Strikingly, these two clinical isolates were now phenotypically resistant to metronidazole (MIC = 8 mg L$^{-1}$ as determined by agar dilution). Ultimately the patient was cured by a second FMT.

We hypothesized that the rCDI episodes were due to clonal RT020 strains that persisted despite antimicrobial therapy and an FMT. Clonal MTZ$^S$ and MTZ$^R$ strains would allow us to determine the underlying genetic changes that resulted in metronidazole resistance. To determine the relatedness between these RT020 isolates whole-genome sequencing (WGS) was performed (Table 1). We also included two more MTZ$^R$ RT020 strains and a non-related RT078 strain isolated from the same patient and 4 MTZ$^S$ and 8 MTZ$^R$ RT010 strains from our laboratory collection (Supplementary Data 1) to perform single-nucleotide polymorphism (SNP) analyses. Strains were considered resistant to metronidazole with MIC values >2 mg L$^{-1}$ according to the EUCAST epidemiological cutoff value[32]. All strains resistant to metronidazole ($n = 12$) showed cross-resistance to the nitroimidazole drug tinidazole.

Assembly of the MTZ$^R$ RT020 strain IB136 (Supplementary Data 2) resulted in a genome of 4166362 bp with 57 contigs, and an average G + C-content of 28.5% (N50 = 263391 bp, mapping rate 98.97%). A BLAST comparison between this genome and the NCBI nt database showed that the genome is closest to the genome of strain LEM1[33]. As expected, 5/6 strains isolated from the patient (all RT020) showed 100% identity over the majority of all contigs, suggesting they are highly similar. All RT020 strains were found to be of multi-locus sequence type (ST) 2, consistent with data from others[34]. The sixth strain (IB137), was a clear outlier and was identified as being closest to the RT078 reference strain M120[35]. This is consistent with another ribotype (RT078) and sequence type (ST11) assignment. All RT010 strains belonged to ST15.

**Resistance does not correlate with a SNP**. Previous studies analyzing the mechanism behind metronidazole resistance in *C. difficile* only studied a single isolate each[29,36]. We performed a core genome SNP analysis on selected strains ($n = 18$; Table 1), comparing MTZ$^S$ ($n = 6$) and MTZ$^R$ ($n = 12$) strains within and between the different PCR ribotypes (RT010, RT020 and RT078).

The evolutionary rate of *C. difficile* has been estimated at 0–2 SNPs per genome per year, but might vary based on intrinsic (strain type) and extrinsic (selective pressure) factors[37]. Our analysis identified a single SNP in the MTZ$^R$ RT020 (IB136), compared to the MTZ$^S$ RT020 strains derived from the same patient, conclusively demonstrating that these strains are clonal. Considering the time of isolation of the susceptible and resistant isolates, this implies the MTZ$^S$ RT020 strain most likely acquired metronidazole resistance. In contrast, between the MTZ$^S$ and MTZ$^R$ RT010 isolates (which come from diverse human and animal sources) 457 SNPs were detected. Moreover, RT010 and RT020 were separated by >25,000 SNPs.

The SNP identified in the RT020 strains discriminating the MTZ$^S$ from the MTZ$^R$ isolates is located in a conserved putative cobalt transporter (CbiN, IPR003705). However, the SNP is not observed in the MTZ$^R$ RT010 strains. Thus, metronidazole resistance is either multifactorial or not contained within the core genome. We did not investigate the contribution of this SNP to metronidazole resistance further.

**MTZ$^R$ *C. difficile* strains contain a 7-kb plasmid**. Next, we investigated extrachromosomal elements (ECEs), which can include plasmids. Although plasmids containing antimicrobial resistance determinants have been described in Gram-positive bacteria, they

appear to be more common in Gram-negatives[38]. Plasmids in *C. difficile* are known to exist, but no phenotypic consequences of plasmid carriage have been described to date[39]. The investigation of the pan-genome of all sequenced strains, including a prediction of ECEs predicted by an in-house pipeline similar to PLACNET[40,41], showed a single contig that was present in all MTZ$^R$ strains (4.6–19.27% of reads mapped, with a minimum of 479497), but absent from MTZ$^S$ strains, of both RT010 and RT020 (0% of reads mapped with a maximum 327 reads). Circularization based on terminal repeats yielded a putative plasmid of 7056 bp with a G + C-content of 41.6% (Fig. 2a). Correct assembly was confirmed by PCR (Fig. 2b) and Sanger sequencing.

To confirm the circular nature of the contig, total DNA isolated from the MTZ$^R$ RT010 strain IB138 before and after PlasmidSafe DNase (PSD, Epicenter)[39] treatment was analyzed by PCR using primers specific for chromosomal DNA (*gluD*) and the putative plasmid (Fig. 2c). A positive signal for *gluD* was only observed in samples that had not been treated with PSD, demonstrating that PSD treatment degrades chromosomal DNA to below the detection limit of the PCR. By contrast, a signal specific for the putative plasmid was visible both before and after PSD treatment. Consequently, we conclude that our whole-genome sequence identified a legitimate 7-kb plasmid.

A total of eight open-reading frames (ORFs) were annotated on the plasmid (Fig. 2a). ORF1-5 encode a hypothetical protein (ORF1), a MobC-like relaxase/Arc-type ribbon-helix-helix (ORF2; PF05713), a MobA/VirD2 family endonuclease relaxase protein (ORF3; PF03432), a hypothetical protein with a MutS2 signature (ORF4), and a predicted replication protein (ORF5), respectively. ORF6 is a small ORF that is likely a pseudogene, and the remaining ORFs encode a metallohydrolase/oxidoreductase protein (ORF7; IPR001279) and a Tn5-like transposase gene (ORF8; PF13701). Intriguingly, ORF6 showed homology on the protein level to the 5-nitroimidazole reductase (*nim*) gene *nimB* (33% identity, 54% positives over 61 amino acids) described in *Bacteroides fragilis* (CAA50578.1) and found in both metronidazole-resistant and susceptible isolates of anaerobic Gram-positive cocci.[42,43] The ORF lacks the region encoding the N-terminal part of the Nim protein, and the Phyre2-predicted protein structure shows it lacks the catalytic site residues. Of note, the plasmid sequences from all strains are highly similar. Compared to the plasmid of strain IB136, only strains IB143, IB144, and IB145 contained a single SNP resulting in a Y286S mutation within the Tn5-like transposase ORF (Supplementary Fig. 1).

Altogether, these results show that all of the MTZ$^R$ strains, but none of the MTZ$^S$ strains, sequenced in this study contain a plasmid, hereafter referred to as pCD-METRO (for plasmid from *C. difficile* associated with metronidazole resistance).

**pCD-METRO is found in strains from different countries**. Two clinical isolates with stable metronidazole resistance have been described and we evaluated the presence of pCD-METRO in the assembled genome sequences from these strains using BLAST[29,36]. We failed to identify pCD-METRO in the draft genome of a toxigenic NAP1 isolate that acquired stable metronidazole resistance through serial passaging under selection[36]. We did identify pCD-METRO (fragmented over multiple contigs) in the draft genome a non-toxigenic Spanish RT010 strain with stable metronidazole resistance (strain 7032989), whereas neither the reduced-susceptible strain nor the susceptible strain from the same study contained the plasmid[29]. We confirmed these results using PCR, as described for strain IB138 (Fig. 3; lanes SP), demonstrating pCD-METRO is indeed present in strain 7032989. These data show that the presence of pCD-METRO may explain

**Table 1 Strains described in this study.**

| Name | Characteristics | PCR ribotype[a] | Toxin profile[b] | MTZ resistance[c] | Source | Reference |
|---|---|---|---|---|---|---|
| 630Δerm | Wild type | 012 | A + B + CDT- | 0.125 (S) | Laboratory | [49] |
| IB125 | 630Δerm pIB86 (pCD-METRO^shuttle); thi^R | 012 | A + B + CDT- | ≥8 (R) | Laboratory | This study |
| IB132 | pCD-METRO− | 020 | A + B + CDT- | 0.25 (S) | Human | This study |
| IB133 | pCD-METRO + | 020 | A + B + CDT- | 8 (R) | Human | This study |
| IB134 | pCD-METRO + | 020 | A + B + CDT- | 8 (R) | Human | This study |
| IB135 | pCD-METRO + | 020 | A + B + CDT- | 8 (R) | Human | This study |
| IB136 | pCD-METRO + | 020 | A + B + CDT- | 8 (R) | Human | This study |
| IB137 | pCD-METRO - | 078 | A + B + CDT+ | 0.125 (S) | Human | This study |
| IB138 | pCD-METRO + | 010 | A- B- CDT- | >8 (R) | Human | This study |
| IB139 | pCD-METRO - | 010 | A- B- CDT- | 1 (S) | Human | This study |
| IB140 | pCD-METRO - | 010 | A- B- CDT- | 0.25 (S) | Human | This study |
| IB141 | pCD-METRO - | 010 | A- B- CDT- | 0.125 (S) | Human | This study |
| IB142 | pCD-METRO - | 010 | A- B- CDT- | 0.125 (S) | Human | This study |
| IB143 | pCD-METRO + | 010 | A- B- CDT- | >8 (R) | Animal | This study |
| IB144 | pCD-METRO + | 010 | A- B- CDT- | >8 (R) | Animal | This study |
| IB145 | pCD-METRO + | 010 | A- B- CDT- | >8 (R) | Animal | This study |
| IB146 | pCD-METRO + | 010 | A- B- CDT- | >8 (R) | Animal | This study |
| IB147 | pCD-METRO + | 010 | A- B- CDT- | >8 (R) | Animal | This study |
| IB148 | pCD-METRO + | 010 | A- B- CDT- | >8 (R) | Animal | This study |
| IB149 | pCD-METRO + | 010 | A- B- CDT- | >8 (R) | Animal | This study |
| IB151 (P016134) | pCD-METRO + | 010 | A- B- CDT- | >8 (R) | Unknown | [81] |
| IB30 | 630Δerm pIB20 (pCD6 replicon, P_{CD0716}-sluc^opt); thi^R | 012 | A + B + CDT- | 0.25 (S) | Laboratory | This study |
| IB90 | 630Δerm pIB80 (pCD-METRO replicon, P_{tet}-gusA); thi^R | 012 | A + B + CDT- | 0.125 (S) | Laboratory | This study |
| LUMCMM18 0002 | pCD-METRO - | 020 | A + B + CDT- | ND (S) | Human | This study |
| LUMCMM19 0348 | pCD-METRO + | 020 | A + B + CDT- | ND (R) | Human | This study |
| LUMCMM19 0830 | pCD-METRO - | 010 | A- B- CDT - | 4 (R) | Unknown | This study |
| LUMCMM19 0880 | pCD-METRO + | 010 | A- B- CDT- | >8 (R) | Unknown | This study |
| LUMCMM19 0970 (7032989) | pCD-METRO + | 010 | A- B- CDT- | >8 (R) | Unknown | [29] |
| LUMCMM19 0960 | pCD-METRO + | 027 | A + B + CDT+ | >8 (R) | Human | [46] |

Listed are strains mentioned in the main body of the manuscript. For a complete overview of all strains used, see Supplementary Data 1.
thi^R thiamphenicol resistance, S susceptible (MIC < 2 mg L$^{-1}$), R resistant (MIC > 2 mg L$^{-1}$) based on the EUCAST epidemiological cutoff for metronidazole[32], ND not determined by agar dilution, but only by E-test (Supplementary Fig. 2).
[a]PCR ribotype determined at the LUMC standard PCR ribotyping.
[b]Toxin profile determined by multiplex PCR.
[c]Metronidazole MIC values in mg L$^{-1}$ as determined by agar dilution conform CLSI guidelines.

at least part of the cases of metronidazole resistance described in literature. We did not detect pCD-METRO in the sequence read archive in entries labeled as *C. difficile*, or otherwise.

Our observations above raise the question how prevalent pCD-METRO is in MTZ^R *C. difficile* isolates and if there is a bias towards specific types or geographic origins. As metronidazole resistance in *C. difficile* is rare, we expanded our collection of clinical isolates through our network (including the ECDC) (n = 76) and with selected strains from the Tolevamer (n = 42) and MODIFY (n = 46) clinical trials[44–46]. To correct for interlaboratory differences in typing and antimicrobial susceptibility testing, all strains were retyped by ribotyping and tested for metronidazole resistance using agar dilution according to Clinical & Laboratory Standards Institute (CLSI) guidelines in our laboratory with inclusion of appropriate control strains[47,48]. Although these strains, with the exception of the Tolevamer strains, were characterized as having altered metronidazole susceptibility by the senders (n = 122), agar dilution performed in our own laboratory classified nearly all of these strains as metronidazole susceptible (MIC < 2 mg L$^{-1}$). We expected pCD-METRO to be present in MTZ^R strains, but not in MTZ^S strains.

We identified three additional metronidazole-resistant strains: a RT027 isolate from Poland (LUMCMM19 0960; MIC > 8 mg L$^{-1}$), a RT010 isolate from the Czech Republic (LUMCMM19 0880; MIC > 8 mg L$^{-1}$) and a RT010 isolate from Germany (P016134;

MIC > 8 mg L$^{-1}$) (Table 1). A PCR on PSD-treated chromosomal DNA isolated from these strains yielded a positive signal using primers targeting the plasmid, but not the chromosome (Fig. 3, lanes PL/CZ/DE), demonstrating all three strains contain pCD-METRO. WGS showed that pCD-METRO in strain LUMCMM19 0960 was identical to that of strain IB136, whereas LUMCMM19 0880 contained a single SNP resulting in a D131N substitution in ORF1 (Supplementary Figure 1). We also screened our laboratory collection of RT010 strains from human and animal sources and identified seven more MTZ^R strains (as determined by both agar dilution and epsilometer tests [E-test]), six of which were positive for pCD-METRO (86%; Supplementary Data 1). A single RT010 strain (LUMCMM19 0830) tested MTZ^R resistant in agar dilution according to CLSI guidelines (MIC = 4 mg L$^{-1}$)[47], but this strain was negative for pCD-METRO in both PCR and WGS. Using E-tests, we found this strain to be susceptible to metronidazole (MIC = 0.19 mg L$^{-1}$) when grown on standard laboratory Brain-Heart Infusion (BHI) agar but resistant (MIC = 16 mg L$^{-1}$) on Brucella Blood Agar (BBA), suggesting a contribution of medium components (possibly heme) to the resistance phenotype (Fig. 4). On both media RT010 control strains IB138 and IB140 are resistant and susceptible, respectively (Fig. 4), with 2–4-fold differences in MIC between the medium conditions. Thus, all pCD-METRO containing strains in this study show medium-independent metronidazole resistance with a

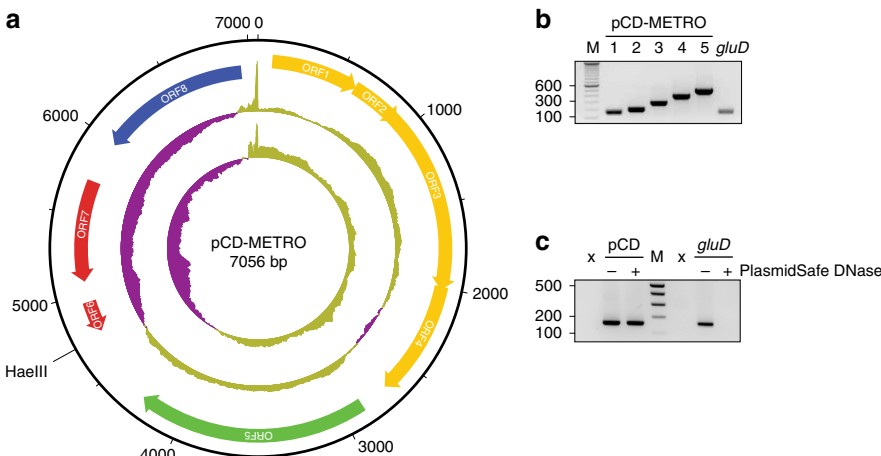

**Fig. 2 pCD-METRO is a 7-kb plasmid. a** Structure of plasmid pCD-METRO and its ORFs. The two innermost circles represent GC content (outer circle) and GC skew (innermost circle) (both step size 5 nt and window size 500nt;, above average in yellow, below average in purple). The unique *Hae*III site used to construct pCD-METRO^shuttle (see methods) is indicated. **b** Gene-specific PCR products amplifying regions of ORFs 6 (lane 1 + 2), ORF5 (lane 3), ORF7 (lane 4) and ORF3 (lane 5), and a chromosomal locus (*gluD*) (**c**) The product of plasmid-specific amplification (targeting ORF6, pCD) or chromosomal-specific amplification (*gluD*) before and after PlasmidSafe DNase treatment. Source data are provided as a Source Data file.

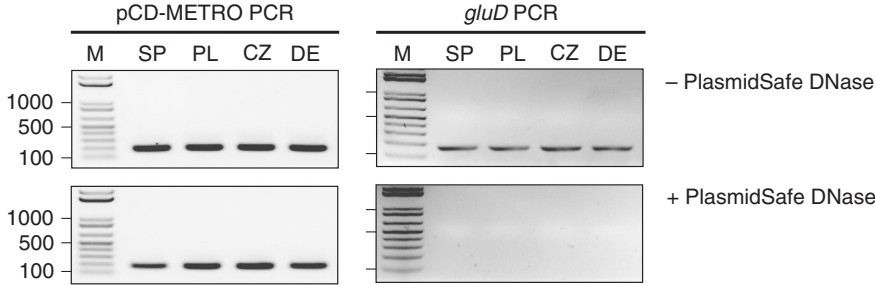

**Fig. 3 pCD-METRO is internationally disseminated.** PCR analysis of strains 7032989 (RT010, Spain) (SP), LUMCMM190960 (RT027, Poland) (PL), LUMCMM19 0880 (RT010, Czech Republic) (CZ), and P016134 (RT010, Germany) (DE). The product of plasmid-specific amplification (targeting ORF8) or chromosomal-specific amplification (*gluD*) before and after PlasmidSafe DNase treatment are shown. Source data are provided as a Source Data file.

$MIC \geq 8\,mg\,L^{-1}$ in agar dilution (22/22). By contrast, all susceptible isolates ($n = 563$) lacked pCD-METRO.

Taken together, our results show that pCD-METRO is internationally disseminated and can explain metronidazole resistance in both non-toxigenic- and toxigenic isolates of *C. difficile*, including those belonging to epidemic ribotypes such as RT027.

**pCD-METRO is likely acquired via horizontal gene transfer**. Our whole-genome sequence analysis suggested the acquisition of pCD-METRO by a toxigenic RT020 strain during treatment of rCDI. We made use of longitudinal fecal samples that were stored during treatment to investigate the presence of pCD-METRO in total fecal DNA at various timepoints. Total DNA derived from the fecal sample harboring the MTZ^S RT020 was positive for the presence of pCD-METRO (Fig. 5). This indicates that pCD-METRO was present in the gut reservoir of the patient. Post-FMT, pCD-METRO was no longer detected in total fecal DNA, suggesting that the fecal transplant reduced levels of pCD-METRO containing *C. difficile* and/or the donor organism to below the limit of detection of the assay. Fecal samples were stored in the absence of cryoprotectant and as a result we were unable to reculture the possible donor organism.

Although we cannot exclude the possibility that the MTZ^R RT020 strain was already present at the moment the MTZ^S

RT020 strain was isolated, our results indicate that pCD-METRO was most likely acquired through horizontal gene transfer between the MTZ^S *C. difficile* strain and an as-of-yet uncharacterized donor organism in the gut of the patient.

**PCR-based identification of metronidazole-resistant strains**. We implemented a PCR targeting pCD-METRO in our routine surveillance and ad hoc typing, as part of the Dutch National Reference Laboratory (NRL) for *C. difficile*. In the period February-August 2019, we characterized 721 strains by ribotyping, and identified a single pCD-METRO-positive strain (LUMCMM19 0348) by PCR. These preliminary data suggest a prevalence of <0.14% in an endemic setting in the Netherlands. The identified strain belonged to RT020 and was confirmed to be MTZ^R in an *E*-test on BBA (Supplementary Fig. 2). As described for the patient case above, we were able to identify an earlier RT020 isolate from the same patient (LUMCMM18 0002) that was pCD-METRO negative and MTZ^S (Supplementary Fig. 2). WGS revealed that the susceptible and resistant strains were identical (0 SNPs difference), but differed as expected in carriage of pCD-METRO. pCD-METRO in the MTZ^R isolate contained two SNPs compared to the plasmid of strain IB136; a G > A conversion located intergenically between ORF6 and ORF7, and a mutation resulting in a V13A mutation in ORF8 (Supplementary Fig. 1).

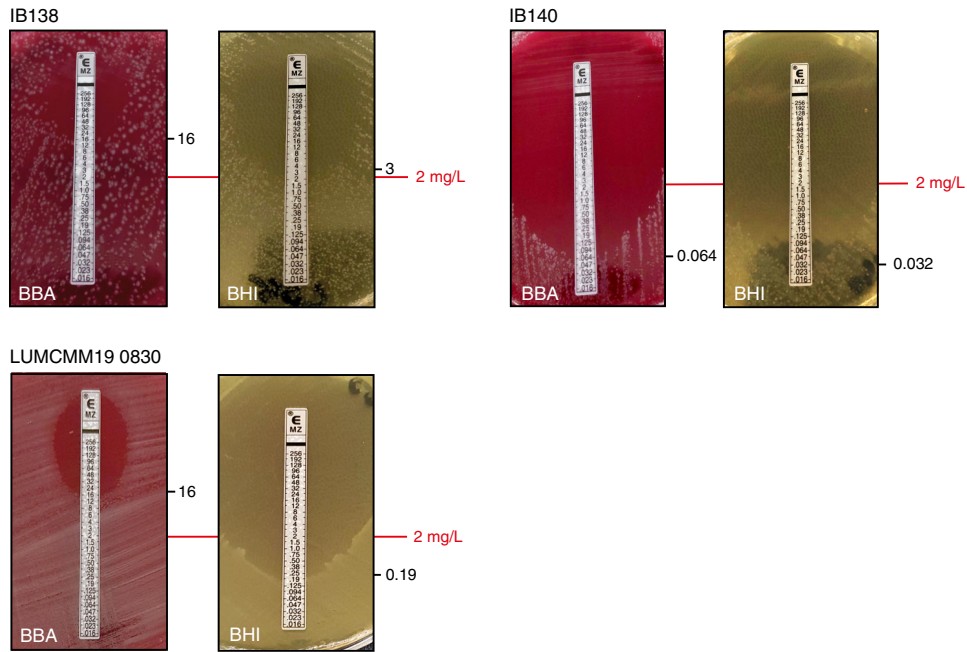

**Fig. 4 Medium-dependent metronidazole resistance.** Strains were grown as described under antimicrobial susceptibility testing in the Methods section and spread onto either Brucella Blood Agar (BBA) plates, or onto BHIY/CDSS agar plates (BHI). E-tests were placed, and plates were incubated for 48 h before imaging. In all, 2 mg L$^{-1}$ is the EUCAST epidemiological cutoff for metronidazole that was used to define resistance in this study[32]. E-test values for the indicated strains are shown next to their respective panels. The images represent three independent repeats with a single replicate per condition. Source data are provided as a Source Data file.

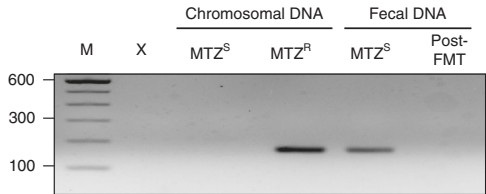

**Fig. 5 pCD-METRO is detectable in fecal total DNA.** pCD-METRO is detectable in fecal total DNA from the same sample from which a MTZ$^S$ RT020 *C. difficile* was isolated. Shown are the results from a PCR targeting ORF6, but similar results were obtained for other plasmid-specific primer sets. Source data are provided as a Source Data file.

Our data suggests that selection by metronidazole is crucial in the acquisition of, or selection for, pCD-METRO containing *C. difficile*.

**pCD-METRO confers metronidazole resistance in *C. difficile*.** Above, we have clearly established a correlation between the presence of pCD-METRO and metronidazole resistance. Next, we sought to unambiguously demonstrate that acquisition of pCD-METRO, and not any secondary events, lead to metronidazole resistance. To generate isogenic strains with or without pCD-METRO, we introduced a shuttle module in the unique *Hae*III restriction site of the plasmid and introduced the resulting vector, pCD-METRO$^{shuttle}$ (pIB86; Supplementary Fig. 3), into the RT012 laboratory strain 630Δ*erm* using standard methods[49]. Metronidazole E-tests showed a reproducible 15-to-20-fold increase in the MIC from 0.064/0.19 mg L$^{-1}$ for the strain without pCD-METRO$^{shuttle}$ to 2–4 mg L$^{-1}$ for the strain with pCD-METRO$^{shuttle}$ (Fig. 6). These results were confirmed using agar dilution, that showed a > 24-fold increase (>5 doubling dilutions) from 0.125-0.25 mg L$^{-1}$ to 8 mg L$^{-1}$ or higher upon introduction of pCD-METRO$^{shuttle}$ (Table 1).

As controls, we included the MTZ$^S$ (IB132) and a MTZ$^R$ (IB133) RT020 strain isolated from the patient. In agreement with the MIC values determined by agar dilution (MIC = 0.25 mg L$^{-1}$ and MIC = 8 mg L$^{-1}$), these isolates showed a MIC corresponding to those observed for the MTZ$^S$ and MTZ$^R$ RT012 isolates, respectively (Fig. 6).

Overall, our results show that acquisition of pCD-METRO is sufficient to raise the MIC of *C. difficile* to values greater than the epidemiological cutoff value defined by EUCAST[32].

**pCD-METRO contains a high copy-number replicon.** Read depth of pCD-METRO in our WGS data indicates an estimated copy number of 100–200, in stark contrast with the pCD6 replicon commonly used in shuttle vectors for *C. difficile* (copy number 4–10)[50]. We wanted to establish the functionality of the predicted replicon and determine the copy number sustained by this replicon in RT012 strains.

A pRPF185-based vector[51] (pIB80) was constructed in which the conventional pCD6 replicon was replaced by a 2-kb DNA fragment of pCD-METRO that includes ORF5, encoding the putative replication protein (Supplementary Fig. 4). Transconjugants containing this vector were readily obtained in the RT012 laboratory strain 630Δ*erm*, demonstrating this region contains a functional replicon.

Next, we compared the relative copy number of the plasmids in overnight cultures by quantitative PCR (qPCR)[50]. Based on the ratio of plasmid-locus *catP* to the chromosomal locus *rpoB*, the copy number of pCD6-replicon vector was ~4, concordant with results of others[50]. By contrast, the copy number of vectors with the pCD-METRO replicon ranges from ~25 (for pIB80, in IB90) to 38 (pCD-METRO$^{shuttle}$, in IB125) (Fig. 7a). We hypothesized that a higher plasmid copy number would also lead to more copies of the resistance marker on the plasmid and thus to a possible increase of resistance to the corresponding antibiotic. Indeed, a strain harboring a *catP*-containing plasmid with the pCD-METRO

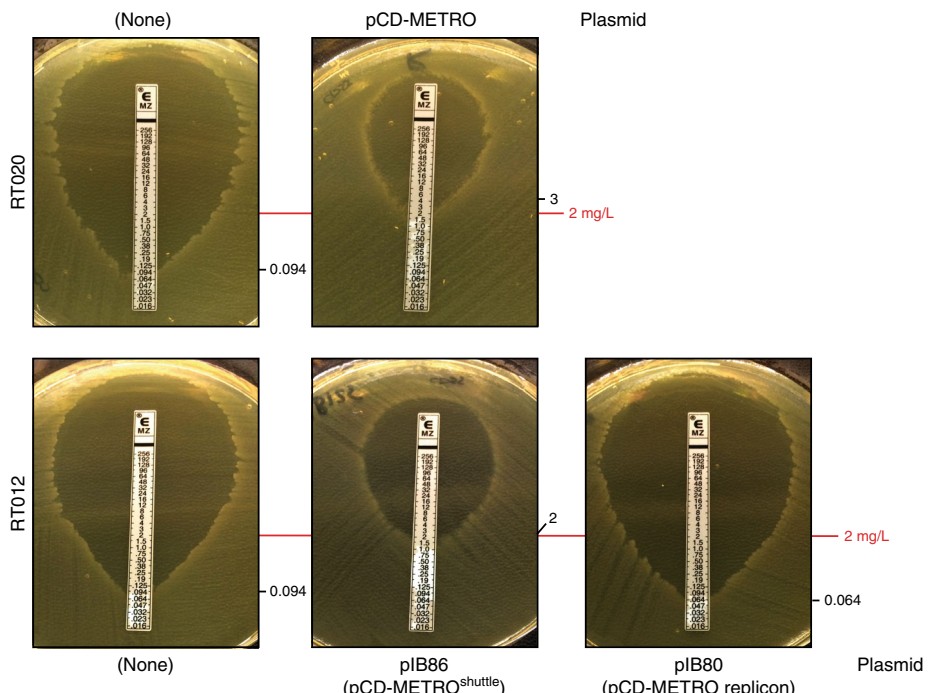

**Fig. 6 pCD-METRO confers metronidazole resistance.** RT020 without plasmid (MTZ$^S$, strain IB132), RT020 with pCD-METRO (MTZ$^R$, strain IB133), RT012 without plasmid (MTZ$^S$, strain 630Δ$erm$), RT012 with pIB86 (pCD-METRO$^{shuttle}$, MTZ$^R$, strain IB125), RT012 with pIB80 (MTZ$^S$, IB90; pIB80 contains the pCD-METRO replicon but lacks the other ORFs of pCD-METRO). IB90 and IB125 are 630Δ$erm$-derivatives[49]. E-tests were performed on BHI agar plates with CDSS. Identical results were obtained on plates without CDSS. The images represent three independent repeats with a single replicate per condition. In all, 2 mg L$^{-1}$ indicates the EUCAST epidemiological cutoff for metronidazole that was used to define resistance in this study[32]. E-test values for the indicated strains are shown next to their respective panels. Source data are provided as a Source Data file.

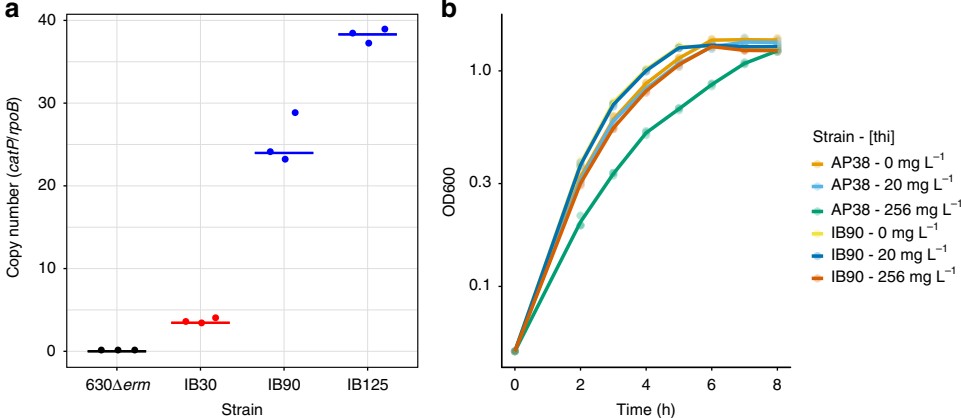

**Fig. 7 The pCD-METRO replicon sustains a high plasmid copy number. a** 630Δ$erm$ is the wild type RT012 laboratory strain. IB30: 630Δ$erm$ + pIB20 (contains pCD6 replicon); IB90: 630Δ$erm$ + pIB80 (contains pCD-METRO replicon); IB125: 630Δ$erm$ + pCD-METRO$^{shuttle}$ (pIB86, contains pCD-METRO replicon). Copy number is determined as the ratio of a plasmid locus (catP) relative to a chromosomal locus (rpoB) as determined by qPCR on total DNA. Data from strains containing a plasmid with the pCD6-replicon are indicated in blue, data from strains containing a plasmid with the pCD-METRO replicon are indicated in red. Individual data points are plotted using symbols. Horizontal lines indicate median values. 630Δ$erm$ vs IB30: $P = 0.1018$; IB90 vs. IB125: $P = 0.0002$; other comparisons: $P < 0.0001$ as determined by two-sided ANOVA and Tukey's test. Experiments were performed in triplicate on three different technical replicates. **b** A strain derived from the laboratory RT012 strain 630Δ$erm$ harboring a plasmid containing the pCD-METRO replicon (IB90) has a growth advantage over a strain containing a plasmid with the pCD6 replicon (AP38) when cultured at high levels of thiamphenicol (thi). AP38: 630Δ$erm$ + pAP24 (pCD6 replicon); IB90: 630Δ$erm$ + pIB80 (pCD-METRO replicon). Growth is measured as an increase in optical density at 600 nm (OD600). Solid line indicates the mean of the individual measurements that are shown using dots (n = 3 biologically independent samples). Source data are provided as a Source Data file.

replicon demonstrates a growth advantage over a strain harboring a similar plasmid with the pCD6 replicon when exposed to high levels (256 mg L$^{-1}$) of thiamphenicol. No significant difference in growth was observed at low concentrations (20 mg L$^{-1}$) of thiamphenicol (Fig. 7b). As pIB80 containing strains are not MTZ$^R$ (Fig. 6), resistance to metronidazole is not mediated by a higher copy number plasmid per se, but is dependent on a determinant specific to pCD-METRO.

A difference between the read-depth estimate and the qPCR can be explained by technical bias or differences in strain background. Nevertheless, our experiments clearly demonstrate that the pCD-METRO replicon sustains plasmid levels that are ~10-fold greater than that of currently used replicons.

We investigated whether the relatively high copy number of pCD-METRO imposes a metabolic cost, by evaluating the growth of strains with and without plasmid in the absence or presence of varying concentrations of metronidazole (Fig. 8). In the absence of metronidazole, the growth of susceptible and resistant strains of both RT012 and RT020 is indistinguishable (Fig. 8a). This was not due to loss of pCD-METRO from the resistant strain, as all colonies tested after the growth experiment had retained the plasmid. With increasing amounts of metronidazole, susceptible strains show a clear growth defect already at the lowest concentration of metronidazole tested ($0.125\,mg\,L^{-1}$), whereas resistant strains do not markedly differ in growth from the control culture at concentrations below the epidemiological cutoff ($2\,mg\,L^{-1}$) (Fig. 8b, c). These values are in agreement with the E-tests performed on the same media (Fig. 6). We conclude that carriage of the plasmid does not affect growth rate in the absence of metronidazole, despite the high copy number, and confers a clear growth advantage in the presence of metronidazole.

We attempted to cure metronidazole-resistant strains of pCD-METRO using serial passaging on non-selective liquid or solid medium. Despite our efforts (Supplementary Methods), we failed to obtain colonies that lacked pCD-METRO, even after non-selective culturing for >50 generations. We hypothesize that the high copy number of pCD-METRO contributes to its stability.

Altogether, these results demonstrate that pCD-METRO encodes a functional replicon that is responsible for a high copy number in *C. difficile* and is efficiently maintained in the absence of selection.

## Discussion

In this study, we describe a plasmid linked to resistance against a clinically relevant antimicrobial in *C. difficile*. We show that the high copy number plasmid pCD-METRO is internationally disseminated, present in diverse PCR ribotypes—including those known to cause outbreaks—and we provide evidence for the possible horizontal transmission of the plasmid. Our data suggests a possible prevalence of <0.14% (1/721; endemic) to 3.9% (22/563; collection enriched for metronidazole-resistant strains) of the plasmid, in line with previous observations[52].

Although the presence of plasmids in *C. difficile* has been known for many years, no phenotypes associated with plasmid carriage have been described[39,41,53]. We show that introduction of pCD-METRO in susceptible strains leads to stable and medium-independent metronidazole resistance. Plasmids may play a broader role in antimicrobial resistance of *C. difficile*. A putative plasmid containing the aminoglycoside/linezolid resistance gene *cfrC* was recently identified in silico, but in contrast to our work no experiments were presented to verify the contig was in fact a plasmid conferring resistance[54]. The presence of an antimicrobial resistance gene does not always result in resistance, and DNA-based identification of putative resistance genes without phenotypic confirmation may lead to an overestimation of the resistance frequencies[19,55,56].

At present, it is unknown which gene(s) on pCD-METRO are responsible for metronidazole resistance. Nitroimidazole reductase (*nim*) genes have been implicated in resistance to nitroimidazole type antibiotics[27]. Although the presence of a truncated *nim* gene on pCD-METRO is intriguing, we do not believe this gene to be responsible for the phenotype for several reasons. Structural modeling of the predicted protein shows that it lacks the catalytic domain, and introduction of the ORF under the

control of an inducible promoter (Supplementary Data 1 and Supplementary Table 2) did not confer resistance in our laboratory strain. Moreover, the RT027 strain R20291 encodes a putative 5-nitroimidazole reductase (R20291_1308) and is not resistant to metronidazole, implying the presence of a *nim* gene is not causally related to metronidazole resistance in *C. difficile* as also noted by others[27]. Further research is necessary to determine the mechanism for metronidazole resistance in *C. difficile* conferred by pCD-METRO, and to investigate the contribution of the high copy number (Fig. 6) to the resistance phenotype.

Our work, combined with that of others, suggests that metronidazole resistance is multifactorial and other factors than pCD-METRO can cause or contribute to metronidazole resistance in *C. difficile*. For instance, pCD-METRO may not explain low level resistance, heterogeneous resistance, or stable resistance resulting from serial passaging of isolated strains under metronidazole selection[27,29,36,57]. We also observed that absolute MIC values in agar dilution experiments differed between MTZ$^R$ isolates of different PCR ribotypes despite carriage of pCD-METRO, suggesting a contribution of chromosomal or other extrachromosomal loci to absolute resistance levels. Although the SNP we identified in the RT020 strain IB136 was not found in other MTZ$^R$ strains of RT010/RT020/RT027, we cannot exclude that it contributes to the resistance in this particular strain. We also observed strong medium-dependent effects: the MICs obtained on BBA are generally higher than those on BHI (Fig. 4), underscoring the importance of using standard conditions for susceptibility testing. Notably, for at least one RT010 strain (LUMCMM19 0830) this led to conversion of the resistance phenotype. Clearly, medium components (possibly iron or heme) contribute to metronidazole resistance. This is in line with suggested metabolic changes in MTZ$^R$ strains that do not harbor pCD-METRO[29,30,36].

The pCD-METRO plasmid is internationally disseminated (Table 1 and Fig. 3), although further research is necessary to determine how prevalent the plasmid is in metronidazole-resistant *C. difficile* isolates. This study attempted to enrich for metronidazole-resistant strains as this resistance is scarce in *C. difficile*. We received strains that were reported to be metronidazole-resistant by the senders. However, when performing antimicrobial susceptibility testing for these strains with agar dilution in our own laboratory, virtually all strains had MIC values below the epidemiological cutoff value from EUCAST for metronidazole and were considered susceptible. For this reason, we ended up having very few metronidazole-resistant isolates of other PCR ribotypes than RT010 (RT020 and RT027). It is not entirely clear how these differences came into existence. Depending on handling of the sample material and freeze-thawing cycles, it is possible that inducible metronidazole resistance, unrelated to pCD-METRO, was initially measured and that this was lost after storage and lack of selection[36]. Considering the apparent stability of pCD-METRO, we do not think that the discrepancies are due to loss of the plasmid during passaging on non-selective media.

Based on the apparent stability of the pCD-METRO plasmid, we consider acquisition rather than loss of pCD-METRO a more likely explanation for the clonality of the RT020 patient isolates. As *C. difficile* has not been demonstrated to be naturally competent for transformation, this suggests that pCD-METRO may be transmissible. Horizontal gene transfer is consistent with the observed level of sequence conservation between the RT010, RT020, and RT027 pCD-METRO plasmids sequenced in this study (Supplementary Fig. 1). Nevertheless, we failed to demonstrate intraspecies transfer with different donor and recipient strains of *C. difficile* under laboratory conditions (Supplementary Methods), suggesting that the strains tested (or possibly the species) lack a determinant required for transfer. Together with its size and the presence of mobilization genes (Fig. 2a), we

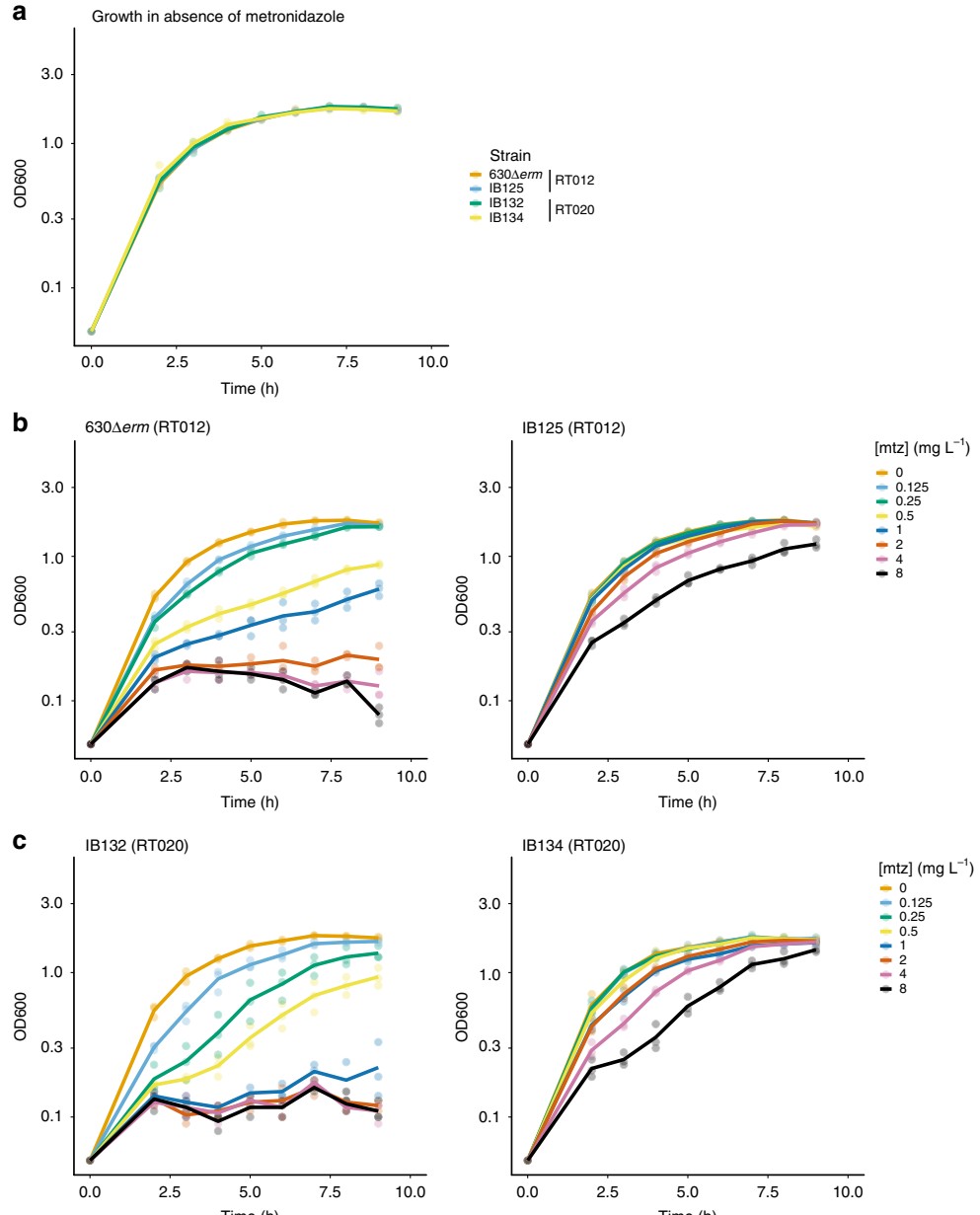

**Fig. 8 The effects of carriage of pCD-METRO on growth. a** Growth as monitored by an increase in optical density at 600 nm (OD600) of various strains in the absence of metronidazole. **b** Growth of RT012 strains in the presence of various concentrations of metronidazole [mtz]. **c** Growth of RT020 strains in the presence of various concentrations of metronidazole [mtz]. Individual measurements are shown as semi-transparent dots. The median of the replicates is shown as a solid line. Source data are provided as a Source Data file.

therefore hypothesize that pCD-METRO is mobilizable from an uncharacterized donor organism[58]. We screened the complete sequence read archive of the NCBI (paired-end Illumina data) for potential sources of the plasmid, but failed to identify any entries with reliable mapping (>1% of data) to pCD-METRO.

As more reports are published associating metronidazole with higher treatment failure[31], a shift in consensus for using metronidazole as first-line treatment for mild to moderate CDI is occurring[59]. The reason for treatment failure is currently unknown, but no correlation between MTZ$^R$ *C. difficile* isolates and treatment failure seems to exist.[55] We also observed that clinical isolates from subjects in which metronidazole treatment failed, were metronidazole susceptible and pCD-METRO negative (Supplementary Data 1)[45]. These observations, however, do not rule out a role for (other) metronidazole-resistant organisms,

potentially harboring pCD-METRO, in treatment failure. Indeed, levels of metronidazole at the end of the colon and in fecal material are low (most likely due to absorption of the drug in the small intestine in the absence of diarrhea)[31], and members of the microbiota involved in inactivation or sequestering of metronidazole may allow for growth of MTZ$^S$ species[60–63].

Our observation of a putatively transmissible plasmid associated with metronidazole resistance in *C. difficile* and the gut microbiome has implications for clinical practice. First, it warrants a further investigation into the role of the plasmid in metronidazole treatment failure in CDI, and—more broadly—in metronidazole resistance of organisms other than *C. difficile*. Second, though this work can be seen as one more argument against the use of metronidazole as a first-line treatment of CDI, detection of the plasmid in fecal material might also guide treatment decisions

(i.e., pCD-METRO harboring patients are excluded from metronidazole treatment). And finally, screening by PCR of fecal donor material intended for FMT might be desirable to reduce the possibility of transferring pCD-METRO from hitherto uncharacterized donor organisms to *C. difficile* in patients.

## Methods

**Strains**. The strains sequenced as part of this study come from various sources. Twenty-one strains were isolated from a single patient at the Leiden University Medical Center (LUMC) or derived from the collection of human and animal isolates of the Dutch NRL for *C. difficile*, which is hosted at the LUMC. Informed consent (approved by the Medical Ethical Committee of the LUMC) was given for the use of the patient samples for research purposes. Other clinical isolates (*n* = 567) were obtained through the NRL and partners in the *C. difficile* typing network of the European Center for Disease Prevention and Control (ECDC), or were previously collected as part of the ECDIS study and the Tolevamer and MODIFY I + II clinical trials[44–46]—two of these were also sequenced. Strain IB136, for which the genome sequence is available from the European Nucleotide Archive (accession CAADHH010000000) has been deposited in the National Collection of Type Cultures (NCTC14385) of Public Health England.

**Whole-genome sequencing and analysis**. DNA was extracted from 9 mL of stationary growth phase cultures grown in BHI (Oxoid) broth using a QIA-symphony (Qiagen, The Netherlands) with the QIAsymphony DSP Virus/Pathogen Midi Kit according to the manufacturer's instructions. All samples were sequenced on an Illumina HiSeq4000 (all samples except those mentioned hereafter) or NovaSeq (LUMCMM18 0002, LUMCMM19 0348, LUMCMM19 0880 and LUMCMM19 0960) platform with read length 150 bp in paired-end mode. All *C. difficile* samples isolated from the patient were assembled using an in-house pipeline, which includes various QC and comparative measures, assembly with six different assemblers as well as scaffolding, but which are not used for all steps and assemblers. Ultimately, Edena v3.131028 was used on the non-trimmed reads with an overlap range between 76 and 146[64]. Reads were mapped back to all assemblies for quality control purposes with Bowtie2 v2.3.1, and SAM files were converted to sorted and indexed BAM files with Samtools v1.5 to obtain mapping rates to the assembly[65,66]. After this step, all contigs from assemblies with a length smaller than 304 bp were discarded, as well as contigs corresponding to the phiX phage spike in (GenBank accession number J02482.1). To remove contaminating contigs, contigs from all assemblies were compared with Blastn v2.6 against the NCBI database (download 10 July, 2018, standard parameters, except e-value of 0.0001)[67,68]. Taxonomy was estimated with the Lowest Common Ancestor algorithm as implemented in MEGAN, except that only Blast matches with a minimum length of 100 bp, and as well only matches not deviating more than 10% in length from the longest match were considered[69]. Filtering was performed on phylum level and the dominant phylum was determined by the amount of base pairs in the assigned blast matches. All contigs assigned to another phylum were discarded. For quality control, the expected genome size was estimated with kmerspectrumanalyzer download August 2013 and Jellyfish v1.1.11[70,71]. Using bedtools genomecov v2.2.16 the read coverage of the assemblies were calculated[72]. All sequence ranges larger than 20 bp with less than 50% coverage and with a larger distance than 200 bp from the contig end were manually inspected, unless only Ns were contained in the sequence. Final evaluation was performed with the values for N50, assembly size (in relation to predicted genome size), mapping rate and manual inspection of low coverage sites. The assembly being evaluated as being best was performed with the Edena assembler and an overlap of 126 (IB136) or 136 (LUMCMM19 0348), with contamination and length filtering, without additional scaffolding and gapfilling.

Annotation was performed with an in-house pipeline as described before[41]. This annotation was furthermore manually reviewed and the annotations from the assembly of *C. difficile* 630 (based on Blastn comparison on gene level) were transferred where applicable.

Extrachromosomal elements were predicted as described before[40,41]. To identify plasmids similar to the pCD-METRO, a homology search was performed with Blastn (with an *e*-value of 0.0001). A further search for non-assembled plasmid sequences was performed. All samples sequenced in paired-end mode on Illumina machines were downloaded from the NCBI with eutils prefetch, and mapped to the plasmid sequence with bowtie2 v2.3.1 to the plasmid sequence. The option–no-mixed was used to supress incorrectly mapping pairs[68].

SNP typing was performed after selecting the best reference assembly. The SNP typing was performed with the in-house pipeline Basty based on the biopet framework[73]. This pipeline performs mapping to the reference assembly with bowtie2 v2.3.1 and SNP typing with BCFtools v1.1-134[74]. Groups were investigated for homozygous SNPs differentiating them. Heterozygous SNPs were discarded. Genomic comparisons between assemblies were performed with Blastn (standard parameters, except an e-value of 0.0001). All programs were executed with standard parameters unless otherwise specified.

Multi-locus sequence type was determined using stringMLST v0.6.2[75] with default settings.

**Antimicrobial susceptibility testing and ribotyping**. All strains were characterized by standardized PCR ribotyping and tested multiple times for metronidazole resistance by agar dilution according to CLSI guidelines, with the inclusion of appropriate microbiological controls[47,48]. No formal breakpoints have been defined for metronidazole resistance in *C. difficile*; here we use the EUCAST epidemiological cutoff of 2 mg L$^{-1}$ to define resistance[32]. Details of all strains and their characteristics are available in Table 1 and Supplementary Data 1. For *E*-tests (BioMerieux), bacterial suspensions corresponding to 1.0 McFarland turbidity were applied on BHI agar supplemented with 0.5% yeast extract (Sigma-Aldrich) and *Clostridium difficile* Selective Supplement (CDSS, Oxoid) or on Brucella Blood Agar plates without antimicrobials. MIC values were read after 48 h of incubation as recommended by CLSI[47].

**Molecular biology techniques**. *Escherichia coli* was cultured aerobically at 37 °C in Luria–Bertani (LB) broth, supplemented with 20 mg L$^{-1}$ chloramphenicol and 50 mg L$^{-1}$ kanamycin when appropriate. *C. difficile* was cultured in BHI supplemented with 0.5% yeast extract, CDSS and 20 mg L$^{-1}$ thiamphenicol when appropriate, in a Don Whitley VA-1000 workstation (10% CO$_2$, 10% H$_2$ and 80% N$_2$ atmosphere).

Plasmids and oligonucleotides are listed in Supplementary Tables 1 and 2, respectively. pIB86 (pCD-METRO$^{shuttle}$, Supplementary Fig. 3 and Supplementary File 1) was constructed using Gibson assembly using *Hae*III-linearized pCD-METRO and a fragment from pRPF185 (Addgene 106367)[51]. This fragment was obtained by PCR, and contained the requirements for maintenance in, and transfer from, *E. coli*. Cesium chloride purified pCD-METRO (see below) was linearized using restriction endonuclease *Hae*III. Primers oWKS-1663 and oWKS-1664 annealed on pRPF185 generating a PCR shuttle-fragment containing pBR322ori-catP-oriT-traJ. To assemble pIB86, 100 ng of insert was assembled against a fourfold molar excess of linearized pCD-METRO backbone using a homemade Gibson Assembly Master Mix (4 U μL$^{-1}$ Taq Ligase (Westburg), 0.004 U μL$^{-1}$ T5 exonuclease (New England Biolabs), 0.025 U μL$^{-1}$ Phusion polymerase (Bioké), 5% polyethyleneglycol (PEG-8000), 10 mM MgCl2, 100 mM Tris-Cl pH = 7.5, 10 mM dithiothreitol, 0.2 mM dATP, 0.2 mM dTTP, 0.2 mM dCTP, 0.2 mM dGTP, and 1 mM β-nicotinamide adenine dinucleotide) for 30 min at 50 °C and transformed into MDS42 cells[76,77]. Transformants were screened by colony PCR using primers oBH-5 and oWKS-1387. The entire sequence of pIB86 was verified by Sanger sequencing using primers oBH-1, oBH-5, oBH-6, oBH-8, oBH-9, oBH-10, oBH-11, oBH-12, oIB-120-, oIB-121, and oIB-122, oWKS-1241-, oWKS-1383, oWKS-1388, oWKS-1537, oWKS-1539, oWKS-1540, oWKS-1574, oWKS-1656, oWKS1658, oWKS-1659, oWKS-1661, oWKS-1663, oWKS-1664 and oWKS-1678. Plasmid pIB80 (Supplementary Fig. 4 and Supplementary File 2) was constructed by ATUM (Newark, CA) and contains a pCD-METRO derived fragment inserted in between the *Kpn*I and *Nco*I sites of pRPF185[51].

Transfer of plasmids from *E. coli* CA434 to *C. difficile* 630Δ*erm*[49] was done using standard methods[78]. Routine DNA extractions were performed using the Nucleospin Plasmid Easypure (Macherey-Nagel) and DNeasy Blood and Tissue (Qiagen) kits after incubating the cells in an enzymatic lysis buffer according to instructions of the manufacturers.

**Isolation of cloning-grade pCD-METRO**. Plasmid pCD-METRO was extracted from 400 mL of culture containing the MTZ$^R$ strain IB138 using the Macherey-Nagel Nucleobond Xtra Midi kit. Using the CsCl$_2$ plasmid purification method this plasmid prep was further cleaned as summarized hereafter. pCD-METRO plasmid prep was added to TE buffer (10 mM Tris pH = 8.0, 1 mM EDTA) and CsCl$_2$ was added to a density of 1 g g$^{-1}$. Approximately 220 μg mL$^{-1}$ ethidium bromide was added to this solution after which samples were spun down in a Beckman Coulter Optima XE-90 ultracentrifuge for 17 h at 65,000 rpm, 20 °C. Bands were visualized with ultraviolet (UV) light and plasmid DNA was collected by withdrawing the lowest of the two resulting bands with an 18 gauge needle. To remove ethidium bromide 1x vol/vol 5 M NaCl saturated N-butanol was used to remove the upper (purple) phase after centrifugation. Samples were ethanol precipitated twice prior to resuspending purified plasmid DNA in TE buffer.

**Plasmid copy number determination**. Real-time quantitative PCR (qPCR) experiments were performed essentially as described[50]. In short, total DNA was isolated after 17 h of growth using a phenol-chloroform extraction protocol and diluted to a concentration of 10 ng μL$^{-1}$. Four microliters of the diluted DNA sample was added to 6 μL of a mixture containing SYBR Green Supermix (Bio-Rad) and gene-specific primers (0.4 μM total) for a total volume of 10 μL per well. Gene-specific primers used were targeting *rpoB* (chromosome) and *catR* (plasmid) and copy number was calculated using the Δ$C_T$ method. Experiments were performed in triplicates on three different technical replicates. Statistical significance was calculated using two-way analysis of variance (ANOVA) and Tukey's test for multiple comparisons (Prism 8, GraphPad)(Supplementary Table 3).

**Preparation of figures**. Agarose gel and *E*-test images were acquired using a Bio-Rad GelDoc XR, processed in Adobe Photoshop CC 2018 (Adobe). Plasmid maps were generated using Artemis and DNAplotter[79,80]. Graphs were generated using PlotTwist (https://huygens.science.uva.nl/PlotTwist/) and PlotsOfData

(https://huygens.science.uva.nl/PlotsOfData/). All figures were prepared for publication in Adobe Illustrator CC 2018 (Adobe).

**Reporting summary**. Further information on research design is available in the Nature Research Reporting Summary linked to this article.

## Data availability

Sequence data that support the findings of this study are available in the European Nucleotide Archive under BioProject number PRJEB24167 with accession numbers ERR2232520-ERR2232537, ERR3611150-ERR3611153, and ERR3772426. The annotated genome assembly for IB136, including pCD-METRO, can be found under accession number CAADHH010000000 and as Supplementary Data 2. The source data underlying Figs. 1, 2b, c, 3–8 and Supplementary Fig. 2 are provided as a Source Data file.

## Code availability

Computer code related to the analysis from this paper is based on published tools, as described in the Methods, and details are available from the authors on request.

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

## Acknowledgements

This work was supported, in part, by a Vidi fellowship (864.13.003) from the Netherlands Organization for Scientific Research, a Gisela Thier Fellowship from the Leiden University Medical Center, and intramural funds to WKS. We would like to thank P. Spigaglia, M. Krutova, R. Peetso, M. Patyi, E. Nováková, E. Piepenbrock, S. Johnson and MSD for strains, and the ECDC for supporting the typing of MTZ[R] strains. We would also like to thank P. Bredenbeek for help in constructing pCD-METRO[shuttle].

## Author contributions

Performed experiments: I.M.B., E.S., C.H., I.M.J.G.B.S., J.C., W.K.S. Analyzed data: B.V.H.H., I.M.B., W.K.S., J.C. Contributed patient samples and metadata: E.M.T., E.J.K. Contributed reagents: R.B., B.V.H.H. Drafted manuscript: I.M.B., B.V.H.H., J.C., E.J.K., W.K.S. All authors edited and approved the final version of the manuscript. The corresponding author had full access to all the data in the study and had final responsibility for the decision to submit for publication.

## Competing interests

W.K.S. has performed research for Cubist and has received speaker fees from Promega. E.J.K. has performed research for Cubist, Novartis, and Qiagen, and has participated in advisory forums of Astellas, Optimer, Actelion, Pfizer, Sanofi Pasteur, and Seres Therapeutics. E.J.K., B.V.H.H., and E.M.T. currently hold an unrestricted research grant from Vedanta Biosciences. These companies had no role in the design and execution of the experiments for this study or the decision to publish. I.M.B., E.S., R.B., C.H., J.C., and I.M.J.G.B.-S.: none to declare. Part of this data has been presented at the International *Clostridium difficile* Symposium 2018, the Scientific Spring Meeting of the KNVM/NVMM 2019, ECCMID 2019, CLOSPATH11 (2019) and Bacterial Morphogenesis, Survival and Virulence: Regulation in 4D (2019).
