## [Peer Review File · Nature Communications]

Reviewers' comments:

Reviewer #1, expert on antibiotics resistance (Remarks to the Author):

Smits et al. Plasmid-mediated metronidazole resistance in *C. difficile*

This paper reports a missing link in the enigma of resistance to nitroimidazole antibiotics. The literature has been plagued with inconsistencies including the lack of a clear role for 5-nitroimidazole reductases (*nim* genes). Accordingly, parameters as diverse as active uptake of nitroimidazoles, expulsion of the antibiotic from the organism, control of nitro-group reduction, capacity to moderate the action of reactive oxidative species and general metabolic fitness of the organism to resist oxidative stress could all contribute to observed resistance.

In relation to infection attributed to *C. difficile*, metronidazole remains an important component of the antimicrobial arsenal with scarce reports of resistance despite many decades of use of this antibiotic. As *C. difficile* infections escalate as a major threat with respect to morbidity and mortality monitoring of resistance to metronidazole becomes important.

The present study is based on a substantial clinical resource and is well executed. Data indicating that the acquisition of the plasmid pCD-METRO confers substantial resistance to metronidazole is solid and convincing. The epidemiological framework for this plasmid remains uncertain because of the low incidence of detection. This aspect is, however, placed correctly in perspective in the description/discussion.

It is disappointing that analysis of the genetic structure of the plasmid has not been informative with respect to resistance to metronidazole. Data presented in the paper did not support intraspecies transfer of the plasmid but rather donation from an unknown organism apparently prevalent in the dysbiotic flora that attends *C. difficile* infections. It was not possible to resolve this important matter.

In summary: an excellent clinical microbiology study that is limited by the absence of mechanism.

Suggestion for improvement

1. Targeting of the plasmid genes to evaluate contribution to resistance
2. Explore parameters of resistance
 - Tagging of metronidazole to assess rate of uptake
 - Examination of metronidazole degradation products by HPLC-Mass Spec
 - Explore metabolic impact of (?) nitro group reduction in resistant organisms by analysis of growth curves in response to titrations of metronidazole

Reviewer #2, clinical expert on *C. diff* (Remarks to the Author):

Boekhoud et al present an elegant analysis of a putatively transmissible plasmid responsible for resistance to metronidazole (MTZ) in multiple *C. difficile* ribotypes. I have two concerns with the paper. 1. The authors have not apparently attempted to cure the plasmid(s) from an isolate and demonstrate loss of MTZ resistance. This would be highly supportive of the plasmid as the source of the resistance and in my view should be done. 2. Failure to demonstrate transmission of the plasmid in *in vitro* experiments makes the claim of "transmissible" tentative and this claim should be moderated in my view. My specific comments are as follows:

Line number

47 Minor point. I believe microbiota is the preferred term rather than flora.

170-172 The lack of a known mechanism for MTZ resistance in the plasmid suggests the possibility of

an as yet unidentified interaction between the plasmid and chromosome to produce resistance. Have the authors identified any novel chromosomal elements common to ribotypes harboring this plasmid and also demonstrating MTZ resistance?

246 How many colonies of RT020 were isolated and examined for presence of MTZ resistance? A mixed population of MTZ resistant and susceptible RT020 cannot be ruled out in my view. The authors have not demonstrated that this plasmid is transferable despite their claim. RT020 plasmid+ was used as a donor in the mating experiments, but was it also tried as a recipient? This is not easy to do unless a unique resistance pattern can be established in the recipient, but this can likely be induced by culture on progressively higher rifampicin to induce *rpoB* mutation as was done for strain EVE17 (Supplement).

267 Increase in MIC from 0.125 to 8 mg/L is a 6-fold increase in doubling dilutions, not 24-fold, I believe.

271 It seems the obvious experiment to demonstrate that pCD-METRO confers metronidazole resistance is to cure the isolate of the plasmid and show that MTZ resistance is no longer present. Have the authors attempted this?

274 Figure 5. It seems the RT020 and RT012 (none) and (pCD-METRO) figures are transposed as the figure lacking plasmid (none) has the bigger zone (higher MIC) on the Etest vs the pCD-METRO figure for both isolates.

325 The evidence that the plasmid is conferring MTZ resistance would be more convincing if the plasmid were cured in a resistant strain and loss of MTZ resistance shown in the absence of the plasmids.

327 Evidence provided of horizontal transmission could also be interpreted as presence of a mixed infection not detected by culture techniques. Suggest that in the absence of demonstrated transmissible resistance that the authors term this "possible transmissible resistance".

374-375 Until transfer can be shown I do not agree there is evidence of horizontal transfer at this time and the authors should moderate this claim.

382 I do not believe higher rates of recurrence with MTZ (compared to vancomycin) has been shown clinically, but significantly decreased cure rates have. Johnson S et al Clin Infect Dis 2014;59:345-54 Suggest the recurrence statement be removed unless a convincing reference can be cited.

390 The reason for MTZ low levels in stool is most likely due to absorption of the drug in the small intestine in the absence of diarrhea. This absence of MTZ in stool has been demonstrated when trying unsuccessfully to eliminate stool carrier state of *C difficile* using MTZ. Ref Johnson S et al Ann Int Med 1992;117:297-302.

392 Suggest adding "possibly" before transmissible to moderate the transmissible claim.

Reviewer #3, clinical expert on *C.diff* (Remarks to the Author):

This is an interesting manuscript that describes novel findings.

The abstract is not well written.

'Metronidazole is used to treat mild- to moderate *Clostridioides difficile* infections (CDI).' This sentence is partially true, but updated guidelines have no longer recommend that metronidazole is used to treat CDI. Thus, the statement requires qualification.

The abstract/study initially appears to be based on a (human) case report. In fact, *C. difficile* isolates from patients with CDI that have reduced susceptibility to metronidazole have been reported on multiple occasions.

The abstract then refers to examining 'Six metronidazole susceptible and 12 metronidazole resistant isolates' before then stating that the pCD-METRO plasmid 'was found in toxigenic and non-toxigenic metronidazole resistant strains from multiple countries (n=22)'. These numbers/denominators do not add up.

The concluding statement 'Our finding that pCD-METRO may be mobilizable can impact diagnostics and treatment of CDI' does not match the findings reported. Why would possible plasmid borne metronidazole affect the diagnosis of CDI?

'Metronidazole is frequently used for the treatment of mild-to-moderate infections and vancomycin for severe infections, though vancomycin is increasingly indicated as a general first-line treatment 9,10 Fidaxomicin has recently also been approved for CDI treatment, but its use is limited by high costs.11' These statements are misleading as the only up-to-date, evidence-graded guideline of the 3 cited references (11) actually states that metronidazole should not be used for the treatment of CDI, and instead recommends either vancomycin or fidaxomicin.

The description of the clinical case of CDI does not accord with Figure 1. For example, the text refers to a single exposure to metronidazole, whereas 3 such exposures appear in Figure 1. The dark green horizontal box in the figure is not defined.

'Previous studies analysing the mechanism behind metronidazole resistance in *C. difficile* only studied one single isolate.23,27'

This statement as worded is not accurate. The sentence refers to two studies that EACH examined one isolate (i.e. 2 in total). Furthermore, other studies (e.g. their reference 16) have examined *C. difficile* isolates in an attempt to determine a mechanism for metronidazole reduced susceptibility.

The authors need to be clear that a formal breakpoint/threshold defining metronidazole resistance in *C. difficile* has not been defined. The (EUCAST) epidemiological cut-off value they are using is not the same as a formal resistance definition.

'This implies the MTZS RT020 strain acquired metronidazole resistance.' The observation could also be due to derepression of metronidazole resistance.

'As metronidazole resistance in *C. difficile* is rare, we expanded our collection of clinical isolates through our network (including the ECDC) and with selected strains from the Tolevamer and MODIFY clinical trials.34-36' Numbers of strains needs to be stated, as it is not clear later in the paragraph the context of the n=122 strains with 'altered metronidazole susceptibility by the senders'. In general the paragraph is not clear and should be rewritten so that the reader can understand the claimed discrepancies. No mention is made of the method used to first describe 122 strains as having reduced susceptibility to metronidazole. Was this testing performed by a reputable laboratory? The storage conditions for these strains is not mentioned. As the authors have elsewhere referred to the instability of metronidazole resistance phenotype, in some instance, this is an important (unaddressed issue).

'A single RT010 strain (LUMCMM19 0830) tested MTZR resistant in agar dilution (MIC=4mg/L), but this strain was negative for pCD-METRO. Thus, all MTZR strains with an MIC \geq 8mg/mL identified here were found to contain pCD-METRO (22/223 22).' The authors are putting too much significance on a one double-dilution difference in MIC. Normally, such a difference would not be considered significant.

This raises a wider question, not addressed in the current manuscript, about the reproducibility of the reported metronidazole MICs. Is storage of isolates in the presence of metronidazole more likely to result in resistance detection being reproducible?

In this context, the authors have not acknowledged that the detection of metronidazole resistance is known to be method dependent. There is also emerging evidence about the role of haem in this phenotype.

'Though we cannot exclude the possibility that the MTZR RT020 strain was already present at the moment the MTZS RT020 strain was isolated, our results indicate that pCD-METRO was most likely

acquired through horizontal gene transfer between the MTZS *C. difficile* strain and an as-of-yet uncharacterized donor organism in the gut of the patient.' While this is possible, the authors must mention that metronidazole exposure may simply have acted as a selection pressure for low numbers of MET R cells e.g. among a heterogeneous *C. difficile* population to expand and so be detectable.

In Figure 5, '2mg/L' appears in red text (twice); it may not be clear to some readers what this referring to. It would be more helpful to state the actual MIC values that the E-test pictures depict.

The main weakness of the manuscript is that the actual mechanism(s) for metronidazole resistance in *C. difficile* has not been determined. The authors acknowledge that there are likely to be several different factors that can contribute to this phenotype. However, we are left without a clear understanding of these and their relative importance both at a strain level and across *C. difficile* populations. The clinical trial strains they have tested are in some cases more than a decade old, meaning that they have not described the contemporaneous epidemiology of plasmid associated metronidazole resistance.

The authors need to acknowledge that metronidazole is considered to be an inferior antibiotic (on the basis of RCT evidence) for treating CDI, reflecting why it has been removed as a first line option in recent guidelines.

The poor pharmacokinetics of metronidazole (in terms of the low levels achieved in the human colon) should be mentioned earlier in the manuscript, preferably in the Introduction, as this could be highly relevant to the selection of resistant strains.

'At present, it is unknown which gene(s) on pCD-METRO are responsible for metronidazole resistance. Nitroimidazole reductase (nim) genes have been implicated in resistance to nitroimidazole type antibiotics ...' The authors should mention their reference 16 here.

'detection of the plasmid in fecal material might also guide treatment decisions' this is speculative and largely impractical considering how CDI is routinely diagnosed.

'And finally, screening of donors of fecal material intended for FMT might be desirable to reduce the possibility of transferring pCD-METRO to *C. difficile* in patients.' What would you screen for and how? Donors are already screened for *C. difficile*.

Response to reviewers

General statement

We thank the reviewers for their comments on our manuscript (NCOMMS-19-23959-T). We have taken them to heart and addressed them in the revised version of our manuscript, which is also provided as a version in which the changes can be tracked. Additionally, we provide a point-by-point response below (our response is in red).

Reviewer comments

Reviewer #1, expert on antibiotics resistance

This paper reports a missing link in the enigma of resistance to nitroimidazole antibiotics. The literature has been plagued with inconsistencies including the lack of a clear role for 5-nitroimidazole reductases (nim genes). Accordingly, parameters as diverse as active uptake of nitroimidazoles, expulsion of the antibiotic from the organism, control of nitro-group reduction, capacity to moderate the action of reactive oxidative species and general metabolic fitness of the organism to resist oxidative stress could all contribute to observed resistance.

In relation to infection attributed to *C. difficile*, metronidazole remains an important component of the antimicrobial arsenal with scarce reports of resistance despite many decades of use of this antibiotic. As *C. difficile* infections escalate as a major threat with respect to morbidity and mortality monitoring of resistance to metronidazole becomes important.

The present study is based on a substantial clinical resource and is well executed. Data indicating that the acquisition of the plasmid pCD-METRO confers substantial resistance to metronidazole is solid and convincing. The epidemiological framework for this plasmid remains uncertain because of the low incidence of detection. This aspect is, however, placed correctly in perspective in the description/discussion.

It is disappointing that analysis of the genetic structure of the plasmid has not been informative with respect to resistance to metronidazole. Data presented in the paper did not support intraspecies transfer of the plasmid but rather donation from an unknown organism apparently prevalent in the dysbiotic flora that attends *C. difficile* infections. It was not possible to resolve this important **matter**. In summary: an excellent clinical microbiology study that is limited by the absence of mechanism.

We appreciate the positive comments from the reviewer that highlights important aspects of the work: the continued use of metronidazole for treatment of *C. difficile* infections, the strengths of our findings and the relevance of the discussion. We share the reviewer's disappointment that we have been unable to resolve a mechanism of resistance to date (discussed in more detail below).

Suggestion for improvement

1. Targeting of the plasmid genes to evaluate contribution to resistance

2. Explore parameters of resistance

- Tagging of metronidazole to assess rate of uptake

- Examination of metronidazole degradation products by HPLC-Mass Spec

- Explore metabolic impact of (?) nitro group reduction in resistant organisms by analysis of growth curves in response to titrations of metronidazole

Reviewer makes several relevant suggestions for tackling the lack of mechanism that is seen as the sole weakness of the manuscript. We would like to highlight the efforts we have undertaken over the past year or so to address these, to show that we believe it will not be possible to resolve this issue in the context of this manuscript.

We have attempted to investigate the function of the pCD-METRO genes in multiple ways.

First, we introduced ORF7 and ORF8 (in our view top candidates for the resistance genes) on a standard cloning vector, either controlled by an inducible promoter or the putative regulatory regions present on pCD-METRO, into our laboratory strain. These strains did not result in a metronidazole resistant phenotype, in contrast to strains harbouring pCD-METRO^{shuttle}; part of this is mentioned in the Discussion. We now also cloned other regions of the plasmid, but transconjugants containing the plasmids containing these regions did also not show metronidazole resistance. The limitation of these experiments is that high copy number may be necessary for resistance to occur. The rep region of pCD-METRO alone did also not confer resistance (Figure 5), however. Thus, it is possible that a combination of ORFs of the plasmid is necessary to result in resistance.

Complementary to this, we attempted to mutate individual ORFs of pCD-METRO^{shuttle}, using mutagenic PCR. Though we could obtain at least some of the desired mutations, sequencing of the full plasmids has revealed multiple secondary mutations, predominantly in the *C. difficile* rep region. As a result, transconjugants could not be obtained. Our interpretation of these findings is that the pCD-METRO replicon likely has some toxicity in *E. coli*, despite our previous successful cloning of pCD-METRO^{shuttle}. In support of this, our yields of pCD-METRO^{shuttle} are consistently very low and not very pure, from a variety of *E. coli* backgrounds (which could also explain a failure in using *in vitro* transposon mutagenesis, that we now also attempted). Our attempts to resolve this using strains suited for the expression of toxic proteins (such as MDS42 and others) have not resulted in success so far. Together, these efforts lead us to believe that pursuing these experiments in the context of this manuscript is not feasible without compromising the novelty and immediacy of our findings. We indicate in the Discussion that further investigations into the mechanism of resistance are necessary.

Regarding the suggestion to pursue parameters of resistance, we note the following. To our knowledge no labelled metronidazole is commercially available and these experiments (using ¹⁴C-metronidazole) were done by one or very few laboratories in the 1970's and 80's (1-4). To perform this experiment, we would have to set up the synthesis or labelling of the compound, and though very interesting, this is beyond the scope of the present study and would require us to set up collaborations with organic chemistry groups. HPLC detection of metronidazole is primarily performed on pharmaceutical formulations (5), not generally in complex mixtures like cell-lysates (as would be required for this study). The analyses are complicated by the rapid intracellular conversion (by reduction) and the formation of metronidazole conjugates (6). We are interested to pursue this avenue in collaboration with our long-term collaborators at the Center for Proteomics and Metabolomics at the LUMC, but are limited by funding for these expensive experiments.

We did perform growth curves of susceptible and resistant strains in response to titrations of metronidazole, as suggested. This information is now contained in Figure 8, and described in the revised manuscript (P13, L299-310). The data shows that there is no growth defect resulting from carriage of pCD-METRO or pCD-METROshuttle in the absence of metronidazole (panel A), and no significant effect on growth at levels of metronidazole well below the MIC for pCD-METRO carrying strains. We conclude that there does not appear to be a metabolic cost of resistance in this setup. Overall, we are continuing our attempts to elucidate the mechanism of metronidazole resistance conferred by pCD-METRO in follow-up studies and are currently attempting to secure additional funding for these experiments.

Reviewer 2

Boekhoud et al present an elegant analysis of a putatively transmissible plasmid responsible for resistance to metronidazole (MTZ) in multiple *C. difficile* ribotypes. I have two concerns with the paper.

1. The authors have not apparently attempted to cure the plasmid(s) from an isolate and demonstrate loss of MTZ resistance. This would be highly supportive of the plasmid as the source of the resistance and in my view should be done.

We share the reviewer's opinion that this would be a useful experiment. We have now attempted to cure pCD-METRO containing strain by serial passaging in medium without metronidazole. Despite these efforts, using both RT012 and RT020 isolates, we did not obtain any strains that lost the plasmid. Thus, pCD-METRO appears to be efficiently maintained in the absence of selection, consistent with the fact that these strains were isolated from the patient and re-cultured on non-selective media prior to the antimicrobial susceptibility testing. Our data is also consistent with data from other groups that demonstrate that high copy number plasmids are not easily cured (note that pCD-METRO occurs at ~10 times the copy number from other currently known *C. difficile* plasmids). We did not attempt to increase the cure rate using chemical mutagens for fear that these would lead to the introduction of secondary mutations. Finally, as we do not know the mechanism (if any) that ensures faithful segregation into daughter cells we are unable to target such a mechanism specifically. We have added text to the Results section of the revised manuscript to describe these experiments and the interpretation thereof (P13, L311-314).

2. Failure to demonstrate transmission of the plasmid in in vitro experiments makes the claim of "transmissible" tentative and this claim should be moderated in my view.

We agree with the reviewer that interspecies transfer between isolated bacterial strains would constitute definitive proof for transmission, but in the absence of the donor we are unable to perform these experiments. Nevertheless, we feel our evidence strongly supports horizontal acquisition. In particular the clonality of the patient isolates (two separate cases are described now) is unlikely to have occurred if the plasmid was not in its entirety acquired by *C. difficile* (based on the results of the curing attempts described above we feel that loss of the plasmid is not likely). To accommodate the reviewer's point of view, we have reviewed the instances where transmission is discussed and adapted the text where appropriate (see for instance P10, and P15, L322, P17, L374-377) – we find that in the remainder of cases we have already taken care to indicate the uncertainty

("most likely acquired", "appears to be transmissible"). We are currently screening fecal material from various sources for pCD-METRO, and in addition plan to investigate the uptake and transmission in microbiome model systems. These experiments are however outside of the scope of the present study, that focuses on the fact that pCD-METRO confers metronidazole resistance.

L47 Minor point. I believe microbiota is the preferred term rather than flora.

Adjusted as requested.

L170-172 The lack of a known mechanism for MTZ resistance in the plasmid suggests the possibility of an as yet unidentified interaction between the plasmid and chromosome to produce resistance. Have the authors identified any novel chromosomal elements common to ribotypes harboring this plasmid and also demonstrating MTZ resistance?

As indicated in the manuscript, a SNP analysis of the initial patient isolates (RT020) revealed only a single chromosomal polymorphism that was not observed in the other PCR ribotypes that contained pCD-METRO (RT010, but also RT027). A more global analysis of gene content, did not reveal any consistent chromosomal make-up that we could conclusively link to metronidazole resistance. We do however explicitly state in the Discussion of the revised manuscript that we agree with the reviewer that a chromosomal contribution to resistance is likely (P16, L350-353). We note that in the newly identified patient case, we did not reliably identify any SNPs between the susceptible and resistant strain (P11, L245-246).

L246 How many colonies of RT020 were isolated and examined for presence of MTZ resistance? A mixed population of MTZ resistant and susceptible RT020 cannot be ruled out in my view. The authors have not demonstrated that this plasmid is transferable despite their claim. RT020 plasmid+ was used as a donor in the mating experiments, but was it also tried as a recipient? This is not easy to do unless a unique resistance pattern can be established in the recipient, but this can likely be induced by culture on progressively higher rifampicin to induce rpoB mutation as was done for strain EVE17 (Supplement).

We agree with the reviewer that we cannot rule out a possible mixed infection, as already noted in our original manuscript ("Though we cannot exclude the possibility that the MTR RT020 strain was already present at the moment the MTZS RT020 strain was isolated, P10, L232-233); indeed, the diagnostic procedure in most institutes – including ours - does not involve culturing multiple isolates. Nevertheless, our data shows that the susceptible and resistant strains are clonal and the most likely scenario explaining the clonality is acquisition of the plasmid by the susceptible strain. It is not possible to use the pCD-METRO positive strain as a recipient in the conjugation experiments, as there is no way of selecting or screening for acquisition of pCD-METRO.

L267 Increase in MIC from 0.125 to 8 mg/L is a 6-fold increase in doubling dilutions, not 24-fold, I believe.

As we indicate actual MIC values, we stand by our statement that the increase is >24-fold ($8/0.25=24$, $8/0.125$ is 64-fold increased). For clarity, however, we have included a statement now that indicates that this correspond to >5 doubling dilutions difference (P12, L263)

L271 It seems the obvious experiment to demonstrate that pCD-METRO confers metronidazole resistance is to cure the isolate of the plasmid and show that MTZ resistance is no longer present. Have the authors attempted this?

See comment 1 of this reviewer.

L274 Figure 5. It seems the RT020 and RT012 (none) and (pCD-METRO) figures are transposed as the figure lacking plasmid (none) has the bigger zone (higher MIC) on the Etest vs the pCD-METRO figure for both isolates.

A bigger halo in the E-test experiment corresponds to a lower MIC, a more susceptible strain. A smaller halo indicates a more resistant strain, with a higher MIC. We carefully evaluated the Figures and come to the conclusion that the data as depicted is correct, with pCD-METRO-less strains demonstrating higher susceptibility and lower MICs.

L325 The evidence that the plasmid is conferring MTZ resistance would be more convincing if the plasmid were cured in a resistant strain and loss of MTZ resistance shown in the absence of the plasmids.

See comment 1 of this reviewer.

L327 Evidence provided of horizontal transmission could also be interpreted as presence of a mixed infection not detected by culture techniques. Suggest that in the absence of demonstrated transmissible resistance that the authors term this "possible transmissible resistance".

This comments is addressed (see major comment 2).

L374-375 Until transfer can be shown I do not agree there is evidence of horizontal transfer at this time and the authors should moderate this claim.

The line numbers here do not contain statements to this effect. However, the comments have been addressed under L327 and major comment 2.

L382 I do not believe higher rates of recurrence with MTZ (compared to vancomycin) has been shown clinically, but significantly decreased cure rates have. Johnson S et al Clin Infect Dis 2014;59:345-54 Suggest the recurrence statement be removed unless a convincing reference can be cited.

We have added the reference suggested by the reviewer to support the statement on treatment failure. We based the statement on recurrence on Johnson et al 2014 Figure 2(7), but upon closer evaluation we agree that the recurrence rate (though higher) is not statistically different from that of vancomycin. We have therefore removed the statement according to the reviewer's suggestion.

L390 The reason for MTZ low levels in stool is most likely due to absorption of the drug in the small intestine in the absence of diarrhea. This absence of MTZ in stool has been demonstrated when trying unsuccessfully to eliminate stool carrier state of C difficile using MTZ. Ref Johnson S et al Ann Int Med 1992;117:297-302.

We have added a statement to indicate this effect with the reference as suggested by the reviewer.

L392 Suggest adding "possibly" before transmissible to moderate the transmissible claim.

Done as suggested.

Reviewer 3

This is an interesting manuscript that describes novel findings.

The abstract is not well written.

‘Metronidazole is used to treat mild- to moderate *Clostridioides difficile* infections (CDI).’ This sentence is partially true, but updated guidelines have no longer recommend that metronidazole is used to treat CDI. Thus, the statement requires qualification.

We acknowledge, as argued by the reviewer, that treatment guidelines in certain cases no longer indicate metronidazole as first line treatment. However, clinical practise is different from clinical guidelines, and due to low cost, wide availability and antibiotic stewardship (possible prevention of VRE due to high use of vancomycin), in combination with a relatively recent change in guidelines (from 2017), metronidazole is still commonly used. Additionally, certain countries may be faster in implementing the guidelines. Moreover, recommendations may differ per patient group (for instance metronidazole is still indicated in guidelines for specific patient groups (8, 9). We believe therefore that the statement as indicated is accurate and as concise as possible for the abstract that is limited to a maximum of 150 words. We have however, to accommodate the reviewer’s view, qualified the use of metronidazole for CDI more clearly in the revised manuscript, and included the updated IDSA guideline reference and the proposed update of the ESCMID guideline (references 9-16 in the manuscript; P3, L52).

The abstract/study initially appears to be based on a (human) case report. In fact, *C. difficile* isolates from patients with CDI that have reduced susceptibility to metronidazole have been reported on multiple occasions.

The size limitations of the abstract do not allow us to describe data not related to the content of the manuscript. The introduction of our manuscript includes multiple references that acknowledge the prevalence of metronidazole resistance in patient isolates (P3, L58-P4, L75). We have therefore not made any changes to the text.

The abstract then refers to examining ‘Six metronidazole susceptible and 12 metronidazole resistant isolates’ before then stating that the pCD-METRO plasmid ‘was found in toxigenic and non-toxigenic metronidazole resistant strains from multiple countries (n=22)’. These numbers/denominators do not add up.

We apologize for the confusion, but “Six” and “12” refer to the strains that were subjected to whole genome sequencing, as detailed in the Results section and the Materials and Methods. The “n=22” refers to a larger collection of strains that was analysed for the presence of pCD-METRO by PCR. To clarify, we have added “by PCR” to the Abstract and adjusted the numbers to reflect the most recent total number of WGSed strains.

The concluding statement 'Our finding that pCD-METRO may be mobilizable can impact diagnostics and treatment of CDI' does not match the findings reported. Why would possible plasmid borne metronidazole affect the diagnosis of CDI?

The statement does not refer to diagnosis, but diagnostics. Our description of pCD-METRO allows the implementation of molecular diagnostics of possible metronidazole resistance of *C. difficile* that can easily be coupled to existing DNA-based methods (including PCR ribotyping, MLVA and multiplex PCR), rather than the time-consuming and labour-intensive antimicrobial susceptibility testing. As a proof of concept, we now include data where we screened all DNA samples that were submitted for routine diagnostics to the Dutch National Reference Laboratory, hosted at the Leiden University Medical Center. We identified a single pCD-METRO PCR positive strain of RT020 from another patient with rCDI that had previously been treated with metronidazole. As for the case described in the initial manuscript, we could trace back an earlier sample from the initial episode of CDI of this patient, and found that it was PCR negative. In line with our other results, only the pCD-METRO positive strain was resistant against metronidazole (Supplementary Figure 1) and the strains were found to be clonal (0 SNPs difference). We have now added this data to the manuscript (P11, L238-251). Our vision is that, in the long term, any molecular diagnosis of pCD-METRO carriage would be coupled back to the treating physician with a negative advice for the use of metronidazole.

'Metronidazole is frequently used for the treatment of mild-to-moderate infections and vancomycin for severe infections, though vancomycin is increasingly indicated as a general first-line treatment 9,10 Fidaxomicin has recently also been approved for CDI treatment, but its use is limited by high costs.11' These statements are misleading as the only up-to-date, evidence-graded guideline of the 3 cited references (11) actually states that metronidazole should not be used for the treatment of CDI, and instead recommends either vancomycin or fidaxomicin.

We agree that different views exist on the use of metronidazole for *C. difficile*, as also discussed above (reviewer 3, first point). We have removed "frequently". We note however, that ref 9 is still an evidence-graded guideline and at present still the most recent version published by the ESCMID and that certain guidelines indicate the use of metronidazole for certain patients groups and there is substantial evidence for a role of metronidazole in the treatment of mild cases of CDI (8-12). We therefore anticipate that metronidazole will only gradually – if at all – lose its place in the therapeutic arsenal against CDI.

Additionally, metronidazole is frequently used in treatment of for instance parasitic infections and in routine abdominal surgery. Even if metronidazole would disappear over time from the arsenal of recommended drugs for treatment of CDI, our findings may bear direct relevance for resistance in other organisms that arises from these uses. We have more clearly written this potential impact in our revised manuscript (e.g. P4, L68-70).

The description of the clinical case of CDI does not accord with Figure 1. For example, the text refers to a single exposure to metronidazole, whereas 3 such exposures appear in Figure 1. The dark green horizontal box in the figure is not defined.

Due to constraints in space and for reasons for privacy, we are unable to give a full description of the clinical case. We wish to point out though that the manuscripts stated "Two more episodes of CDI occurred during which the patient was treated primarily with vancomycin, prior to an FMT...", which

does not exclude (minor) episodes of metronidazole treatment. Nevertheless, as metronidazole may be relevant for this study, we now make this point explicitly in the revised manuscript (P5, L89-93). We are unsure what the reviewer has in mind when referring to the dark green box, as only light green boxes are shown which correspond to vancomycin treatment, and are indicated in the legend.

'Previous studies analysing the mechanism behind metronidazole resistance in *C. difficile* only studied one single isolate.^{23,27} This statement as worded is not accurate. The sentence refers to two studies that EACH examined one isolate (i.e. 2 in total). Furthermore, other studies (e.g. their reference 16) have examined *C. difficile* isolates in an attempt to determine a mechanism for metronidazole reduced susceptibility.

We have added "each" to the statement (P6, L119-120). In contrast to the reviewer's statement, ref 16 (13) does not investigate the mechanism behind the resistance and therefore bears no direct relevance for this sentence.

The authors need to be clear that a formal breakpoint/threshold defining metronidazole resistance in *C. difficile* has not been defined. The (EUCAST) epidemiological cut-off value they are using is not the same as a formal resistance definition.

We have added a statement to more clearly indicate this fact throughout our revised manuscript (e.g. P5, L106; P12, L270; P16, L367; P20, L432; and the legends of Figure 4, Figure 6 and Table 1).

'This implies the MTZS RT020 strain acquired metronidazole resistance.' The observation could also be due to derepression of metronidazole resistance.

Resistance is a phenotype, that in itself cannot be derepressed; we therefore assume that the reviewer refers to derepression of a gene conferring resistance already present in the strain. We consider this unlikely as a) we find no chromosomal changes and b) the observed phenotypic conversion from susceptible to resistant is stable in the absence of selection. We have addressed the reviewer's comment by rephrased the sentence to read "Considering the time of isolation of the susceptible and resistant isolates, this implies the MTZ^s RT020 strain most likely acquired metronidazole resistance" (P6, L126-127) that allows a possibility of alternative scenario's.

'As metronidazole resistance in *C. difficile* is rare, we expanded our collection of clinical isolates through our network (including the ECDC) and with selected strains from the Tolevamer and MODIFY clinical trials.³⁴⁻³⁶ Numbers of strains needs to be stated, as it is not clear later in the paragraph the context of the n=122 strains with 'altered metronidazole susceptibility by the senders'. In general the paragraph is not clear and should be rewritten so that the reader can understand the claimed discrepancies. No mention is made of the method used to first describe 122 strains as having reduced susceptibility to metronidazole. Was this testing performed by a reputable laboratory? The storage conditions for these strains is not mentioned. As the authors have elsewhere referred to the instability of metronidazole resistance phenotype, in some instance, this is an important (unaddressed issue).

We have added the numbers requested by the reviewer (P9, L186-197). Testing is described in the publications on the TOLEVAMER and MODIFY studies that were cited in the manuscript (7, 14) and was performed by reputable laboratories or organisations. Testing methodology (including culturing) for the strains from our network is unknown, but within the ECDC agar dilution according to CLSI

guidelines (on Brucella Blood agar) is default. Our concerns with the discrepancies are clearly laid out in the Discussion of both the original and Revised manuscript (P16, L369-P17, L373).

'A single RT010 strain (LUMCMM19 0830) tested MTZR resistant in agar dilution (MIC=4mg/L), but this strain was negative for pCD-METRO. Thus, all MTZR strains with an MIC \geq 8mg/mL identified here were found to contain pCD-METRO (22/223 22).' The authors are putting too much significance on a one double-dilution difference in MIC. Normally, such a difference would not be considered significant.

We shared the reviewer's concern for the reproducibility of the agar dilution data. Therefore we performed the assays for these strains multiple times, with the inclusion of appropriate (*C. difficile* and non-*C. difficile*) controls with the same result (we now made this explicit in the Methods, see e.g. P19, L429-431). The result is significant to describe as it is the only strain we identified so far with an MIC above the ECOFF that is pCD-METRO negative. We will provide further support for a different mechanism of resistance below (haem discussion, see also P9, L205-P10, L217).

This raises a wider question, not addressed in the current manuscript, about the reproducibility of the reported metronidazole MICs. Is storage of isolates in the presence of metronidazole more likely to result in resistance detection being reproducible?

As indicated, the assays were performed multiple times for all strains described in the manuscript and showed reproducible results. Our isolates are stored in the absence of metronidazole, and we have not systematically investigated the possible differences in MIC that occur when isolates are stored, or precultured, in metronidazole containing medium. For this reason, we do not discuss this in the manuscript. Our storage and growth conditions were chosen to minimize confounding of inducible metronidazole resistance.

In this context, the authors have not acknowledged that the detection of metronidazole resistance is known to be method dependent. There is also emerging evidence about the role of haem in this phenotype.

We agree that results from antimicrobial susceptibility testing vary per method; this is also evident from the differences in MIC values determined with agar dilution and E-test as described in our manuscript. We therefore have taken care to test all isolates using the golden standard, agar dilution, according to CLSI standards. We thank the reviewer for suggesting an effect of haem. Though we could not find a reference describing haem-dependent metronidazole resistance for *C. difficile*, we did actually observe a possibly haem-dependent difference in resistance with isolate LUMCMM19 0830. This isolate tested resistant in agar dilution using Brucella Blood agar plates, but did not contain pCD-METRO (in contrast to all other strains). Based on this observation, we investigated this further and found that it was susceptible when tested on common laboratory medium using for our work with genetically modified *C. difficile* (BHI). Similar results were obtained with this strain in E-tests on Brucella blood plates versus BHI. In contrast, pCD-METRO containing strains tested resistant with both methods, though also in this case a medium dependent effect was observed. This data has now been added to the revised manuscript (P9, L205-P10, L217; Revised Figure 4). We thus further qualify the contribution of pCD-METRO to metronidazole resistance and add a clear description of different mechanisms of resistance.

'Though we cannot exclude the possibility that the MTZR RT020 strain was already present at the moment the MTZS RT020 strain was isolated, our results indicate that pCD-METRO was most likely acquired through horizontal gene transfer between the MTZS *C. difficile* strain and an as-of-yet uncharacterized donor organism in the gut of the patient.' While this is possible, the authors must mention that metronidazole exposure may simply have acted as a selection pressure for low numbers of MET R cells e.g. among a heterogenous *C. difficile* population to expand and so be detectable.

The possibility of a mixed population is pointed out in the Discussion of the manuscript ("Though we cannot exclude the possibility that the MTR RT020 strain was already present at the moment the MTZS RT020 strain was isolated"). We have now added a statement about the role for positive selection by MTZ, as this is also consistent with the newly included data on the unbiased screen of NRL isolates (P11, L238-251), where we once more isolated a metronidazole resistant RT020 strain from a patient with recurrent CDI that had previously been treated with metronidazole (see above).

In Figure 5, '2mg/L' appears in red text (twice); it may not be clear to some readers what this referring to. It would be more helpful to state the actual MIC values that the E-test pictures depict.

We have added the following statement to all relevant Figure Legends: "2 mg/L indicates the EUCAST ECOFF for metronidazole [ref] that was used to define resistance in this study" and all E-test figures (Figures 4 and 6) now include an indication of MIC values next to each panel.

The main weakness of the manuscript is that the actual mechanism(s) for metronidazole resistance in *C. difficile* has not been determined. The authors acknowledge that there are likely to be several different factors that can contribute to this phenotype. However, we are left without a clear understanding of these and their relative importance both at a strain level and across *C. difficile* populations. The clinical trial strains they have tested are in some cases more than a decade old, meaning that they have not described the contemporaneous epidemiology of plasmid associated metronidazole resistance.

As pointed out in the manuscript, metronidazole resistance in *C. difficile* is relatively rare and the identification of the plasmid involved in metronidazole resistance would probably not have been possible without the use of a (partially) historic collection of strains. We did not aim to describe the current epidemiology. However, we have now added data from our routine surveillance through the National Reference Laboratory for *C. difficile* (as also pointed out in the comment on the concluding remark), hosted at the Leiden University Medical Center. Since starting the routine surveillance for plasmid-mediated metronidazole resistance, we have identified a single positive (and resistant) isolate among ~700 strains. Though these efforts are ongoing and we do not wish to draw conclusions about prevalence of the plasmid, this suggests that ~0.14% of modern strains are metronidazole resistant. This data has been added to the manuscript (P11, L241-242).

The authors need to acknowledge that metronidazole is considered to be an inferior antibiotic (on the basis of RCT evidence) for treating CDI, reflecting why it has been removed as a first line option in recent guidelines.

This point is extensively discussed above and has been addressed in a revised version of the manuscript.

The poor pharmacokinetics of metronidazole (in terms of the low levels achieved in the human colon) should be mentioned earlier in the manuscript, preferably in the Introduction, as this could be highly relevant to the selection of resistant strains.

A statement to this effect has been added to the revised manuscript (P4, L74-75).

‘At present, it is unknown which gene(s) on pCD-METRO are responsible for metronidazole resistance. Nitroimidazole reductase (nim) genes have been implicated in resistance to nitroimidazole type antibiotics ...’ The authors should mention their reference 16 here.

Reference 16 does not contain any pertinent information to the statement made; it mentions only that a single RT010 strain of *C. difficile* was negative for *nim*, and this does not support their implication in nitroimidazole type antibiotics. We therefore made no changes.

‘detection of the plasmid in fecal material might also guide treatment decisions’ this is speculative and largely impractical considering how CDI is routinely diagnosed.

We acknowledge that this is speculative. However, we do not necessarily consider this to be impractical. Typing of *C. difficile* on fecal material or DNA isolated from such material may become feasible in the (near) future. The inclusion of a molecular assay to detect pCD-METRO (e.g. PCR) could be done on the same material, and if positive, could result in a negative advice for treatment with metronidazole. Irrespective of current clinical practise, we feel this is worth pointing out.

‘And finally, screening of donors of fecal material intended for FMT might be desirable to reduce the possibility of transferring pCD-METRO to *C. difficile* in patients.’ What would you screen for and how? Donors are already screened for *C. difficile*.

Similar to the point above, this statement is included to provide an outlook on further developments that can occur after the identification of pCD-METRO. The two cases currently described (the RT020 from the manuscript and the “new” RT020 case that we identified through the National Reference Laboratory) both present cases of rCDI in which the initial episode was treated with metronidazole. FMT is at present indicated for cases of refractory/recurrent CDI. The putative transfer of pCD-METRO appears to occur in this context and it would be beneficial to ensure that a patient with fulminant CDI does not receive donor material that harbors pCD-METRO (irrespective of the species it is contained in) to reduce the spread of this resistance in *C. difficile*. This has been clarified in the last sentence of the manuscript (P18, L397-406).

Editorial comments

- Editorial checklist is uploaded with the revised version of the manuscript
- Reporting summary is uploaded
- Formatting requirements have been taken into account to the best of our ability
- Source data file is now provided containing uncropped images of all gels/plates (Figures 2, 3, 4, 5, 6 and Supplementary Figure 1), as well as raw data depicted in Figure 1, 7 and 8.
- Data availability and code availability statement is included

References cited in Response to reviewers

1. Koch RL, Goldman P. The anaerobic metabolism of metronidazole forms N-(2-hydroxyethyl)-oxamic acid. *J Pharmacol Exp Ther.* 1979;208(3):406-10.
2. Muller M, Gorrell TE. Metabolism and metronidazole uptake in *Trichomonas vaginalis* isolates with different metronidazole susceptibilities. *Antimicrobial agents and chemotherapy.* 1983;24(5):667-73.
3. Muller M, Lindmark DG. Uptake of metronidazole and its effect on viability in trichomonads and *Entamoeba invadens* under anaerobic and aerobic conditions. *Antimicrobial agents and chemotherapy.* 1976;9(4):696-700.
4. Malliaros DP, Goldman P. Interaction of metronidazole with *Escherichia coli* deoxyribonucleic acid. *Biochem Pharmacol.* 1991;42(9):1739-44.
5. Khattab FI, Ramadan NK, Hegazy MA, Ghoniem NS. Simultaneous determination of metronidazole and spiramycin in bulk powder and in tablets using different spectrophotometric techniques. *Drug Test Anal.* 2010;2(1):37-44.
6. Dingsdag SA, Hunter N. Metronidazole: an update on metabolism, structure-cytotoxicity and resistance mechanisms. *The Journal of antimicrobial chemotherapy.* 2018;73(2):265-79.
7. Johnson S, Louie TJ, Gerding DN, Cornely OA, Chasan-Taber S, Fitts D, et al. Vancomycin, metronidazole, or tolevamer for *Clostridium difficile* infection: results from two multinational, randomized, controlled trials. *Clinical infectious diseases : an official publication of the Infectious Diseases Society of America.* 2014;59(3):345-54.
8. Diorio C, Robinson PD, Ammann RA, Castagnola E, Erickson K, Esbenshade A, et al. Guideline for the Management of *Clostridium Difficile* Infection in Children and Adolescents With Cancer and Pediatric Hematopoietic Stem-Cell Transplantation Recipients. *J Clin Oncol.* 2018;JCO1800407.
9. Sartelli M, Di Bella S, McFarland LV, Khanna S, Furuya-Kanamori L, Abuzeid N, et al. 2019 update of the WSES guidelines for management of *Clostridioides (Clostridium) difficile* infection in surgical patients. *World J Emerg Surg.* 2019;14:8.
10. Fabre V, Dzintars K, Avdic E, Cosgrove SE. Role of Metronidazole in Mild *Clostridium difficile* Infections. *Clinical infectious diseases : an official publication of the Infectious Diseases Society of America.* 2018;67(12):1956-8.
11. Stevens VW, Nelson RE, Schwab-Daugherty EM, Khader K, Jones MM, Brown KA, et al. Comparative Effectiveness of Vancomycin and Metronidazole for the Prevention of Recurrence and Death in Patients With *Clostridium difficile* Infection. *JAMA Intern Med.* 2017;177(4):546-53.
12. Appaneal HJ, Caffrey AR, LaPlante KL. What is the role for metronidazole in the treatment of *Clostridium difficile* infection? Results from a national cohort study of Veterans with initial mild disease. *Clinical infectious diseases : an official publication of the Infectious Diseases Society of America.* 2018.
13. Brazier JS, Fawley W, Freeman J, Wilcox MH. Reduced susceptibility of *Clostridium difficile* to metronidazole. *The Journal of antimicrobial chemotherapy.* 2001;48(5):741-2.
14. Wilcox MH, Gerding DN, Poxton IR, Kelly C, Nathan R, Birch T, et al. Bezlotoxumab for Prevention of Recurrent *Clostridium difficile* Infection. *The New England journal of medicine.* 2017;376(4):305-17.

REVIEWERS' COMMENTS:

Reviewer #1 (Remarks to the Author):

The authors have responded to all of my suggestions for improvement. They have openly described difficulties encountered and have detailed the lack of resolution where this has occurred.

It is my opinion that the combination of modifications to the manuscript together with the availability of reviewer comments and responses to the readers of NComm. provides a comprehensive analysis of the subject matter.

The assembled material represents a significant contribution that is appropriate for publication in Nature Communications.

Neil Hunter

Reviewer #2 (Remarks to the Author):

The authors are to be commended for this thorough investigation of possible MTZ plasmid mediated resistance. However, it remains disappointing that,

1. No mechanism has been determined.
2. Plasmid transfer has not been demonstrated.
3. Plasmid curing has not been shown to determine if resistance disappears.

Reviewer #3 (Remarks to the Author):

Thank you for the manuscript revisions in response to my comments.

The revised abstract now starts with 'Metronidazole is used to treat *Clostridioides difficile* (CDI) infections', which remains (possibly even more than originally) misleading. Qualification is needed given that there is high quality evidence that metronidazole is inferior to alternatives, and which has informed recent guidelines to recommend AGAINST the continued use of this agent. The revised text on P3 lines 50-53 does not acknowledge the proven inferiority of metronidazole. The authors state correctly that existent ESCMID guidelines do recommend the use of metronidazole, but fail to acknowledge that these are >5 years old (and so are based on even older data).

The authors speculate that a PCR assay could be used to detect the resistance plasmid in fecal material to guide treatment decisions. Given the low prevalence of the resistance plasmid, the notably poor PK of metronidazole and the clear evidence of inferiority of metronidazole for treating CDI, it is unlikely that the results of such an assay would really alter a clinical decision on whether to use metronidazole. To be clear, I would not be happy to receive metronidazole to treat my CDI even if the theoretical assay found no evidence of the resistance plasmid in a fecal sample.

Response to reviewers

Reviewer #1 (Remarks to the Author)

The authors have responded to all of my suggestions for improvement. They have openly described difficulties encountered and have detailed the lack of resolution where this has occurred.

It is my opinion that the combination of modifications to the manuscript together with the availability of reviewer comments and responses to the readers of NComm. provides a comprehensive analysis of the subject matter.

The assembled material represents a significant contribution that is appropriate for publication in Nature Communications.

Neil Hunter

We thanks prof. Hunter for this assessment.

Reviewer #2 (Remarks to the Author)

The authors are to be commended for this thorough investigation of possible MTZ plasmid mediated resistance. However, it remains disappointing that,

1. No mechanism has been determined.
2. Plasmid transfer has not been demonstrated.
3. Plasmid curing has not been shown to determine if resistance disappears.

We share the reviewer's disappointment, but feel that the manuscript clearly demonstrates a role for pCD-METRO in resistance, demonstrates the clonal nature of the plasmid-less and plasmid containing isolates that differ in metronidazole resistance, and have demonstrated that the high copy number likely prevents plasmid-loss.

Reviewer #3 (Remarks to the Author)

Thank you for the manuscript revisions in response to my comments.

The revised abstract now starts with 'Metronidazole is used to treat *Clostridioides difficile* (CDI) infections', which remains (possibly even more than originally) misleading. Qualification is needed given that there is high quality evidence that metronidazole is inferior to alternatives, and which has informed recent guidelines to recommend AGAINST the continued use of this agent. The revised text on P3 lines 50-53 does not acknowledge the proven inferiority of metronidazole. The authors state correctly that existent ESCMID guidelines do recommend the use of metronidazole, but fail to acknowledge that these are >5 years old (and so are based on even older data).

The abstract has been reconstructed and clearly indicates recent guideline changes now, as suggested by the editor. We have not clarified the rest of the text further, as the point of the manuscript is not to discuss whether or not metronidazole is a suitable drug for treatment of *C. difficile*, but to implicate a plasmid in antimicrobial resistance in *C. difficile*. An elaborate discussion

on the first line treatments would distract from the message, and is better suited for a clinical journal in the authors' opinions.

The authors speculate that a PCR assay could be used to detect the resistance plasmid in fecal material to guide treatment decisions. Given the low prevalence of the resistance plasmid, the notably poor PK of metronidazole and the clear evidence of inferiority of metronidazole for treating CDI, it is unlikely that the results of such an assay would really alter a clinical decision on whether to use metronidazole. To be clear, I would not be happy to receive metronidazole to treat my CDI even if the theoretical assay found no evidence of the resistance plasmid in a fecal sample.

Our experiments clearly show that the PCR reliably detects resistant isolates (Supplemental Figure 2), and this information could potentially be used to recommend against metronidazole in those situations (countries, specific patient groups) where it is still used. We appreciate the fact that reviewer would not opt for metronidazole in treatment for *C. difficile*, and can only hope that patients world-wide will be in a situation where availability and cost of the available drugs will allow them to make this decision themselves.